



# Spatiotemporal variability of light attenuation and net ecosystem metabolism in a back-barrier estuary

Neil K. Ganju[1], Jeremy M. Testa[2], Steven E. Suttles[1], Alfredo L. Aretxabaleta[1]

[1]U.S. Geological Survey, Woods Hole Coastal and Marine Science Center, Woods Hole, MA

[2]Chesapeake Biological Laboratory, University of Maryland Center for Environmental Science, Solomons, MD

*Correspondence to:* Neil K. Ganju (nganju@usgs.gov)

**Abstract.** Quantifying system-wide biogeochemical dynamics and ecosystem metabolism in estuaries is often attempted using a long-term continuous record at a single site, or short-term records at multiple sites due to sampling limitations that preclude long-term monitoring at multiple sites. However, differences in the dominant primary

producer at a given location (e.g., phytoplankton versus submerged aquatic vegetation; SAV) control diel variations in dissolved oxygen and associated ecosystem metabolism, and may confound metabolism estimates that do not account for this variability. We hypothesize that even in shallow, well-mixed estuaries there are strong spatiotemporal gradients in ecosystem metabolism due to the influence of submerged aquatic vegetation (SAV), and ensuing feedbacks to sediment resuspension, light attenuation, and primary production. We tested this hypothesis by

measuring hydrodynamic properties, biogeochemical variables (fluorescent dissolved organic matter (fDOM), turbidity, chlorophyll-a fluorescence, dissolved oxygen), and photosynthetically active radiation (PAR) over one year at 15 min intervals at paired channel (unvegetated) and shoal (vegetated) sites in Chincoteague Bay, Maryland/Virginia, USA, a shallow back-barrier estuary. Light attenuation ($K_{dPAR}$) at all sites was dominated by turbidity from suspended sediment, with lower contributions from fDOM and chlorophyll-a. However, there was

significant seasonal variability in the resuspension-shear stress relationship on the vegetated shoals, but not in adjacent unvegetated channels. This indicated that $K_{dPAR}$ on the shoals was mediated by SAV presence in the summer, which reduced resuspension and therefore $K_{dPAR}$. We also found that gross primary production ($P_g$) and $K_{dPAR}$ were significantly negatively correlated on the shoals and uncorrelated in the channels, indicating that $P_g$ over the vegetated shoals is controlled by a feedback loop between SAV presence, sediment resuspension, and light

availability. Metabolic estimates indicated substantial differences in net ecosystem metabolism between vegetated and unvegetated sites, with the former tending towards net autotrophy in the summer. Ongoing trends of SAV loss





in this and other back-barrier estuaries suggests that these systems may also shift towards net heterotrophy, reducing their effectiveness as long-term carbon sinks. With regard to temporal variability, we found that varying sampling frequency between 15 min and 1 d resulted in comparable mean values of biogeochemical variables, but extreme values were missed by daily sampling. In fact, daily re-sampling minimized the variability between sites and falsely

suggested spatial homogeneity in biogeochemistry, emphasizing the need for high-frequency sampling. This study confirms that properly quantifying ecosystem metabolism and associated biogeochemical variability requires characterization of the diverse estuarine environments, even in well-mixed systems, and demonstrates the deficiencies introduced by infrequent sampling on the interpretation of spatial gradients.

**1 Introduction**

Back-barrier estuaries are biologically productive environments that provide numerous ecological, recreational, and economic benefits. Submerged aquatic vegetation (SAV) proliferates in these environments due to relatively shallow bathymetry and sufficient light availability, providing habitat for many fish and crustaceans (Heck and Orth 1980) as well as enhancing wave attenuation (Nowacki et al. 2017). Primary production in back-barrier estuaries and similar

shallow marine ecosystems is relatively high given the shallow bathymetry, benthic light availability, and sometimes large SAV beds (e.g., Duarte and Chiscano 1999). Benthic communities within shallow ecosystems host other primary producers where SAV are absent, including microphytobenthos (Sundbäck et al. 2000) and various forms of macroalgae, and the relative contribution of these producers is altered by nutrient enrichment (e.g., McGlathery 2001, Valiela et al. 1997). In deeper, unvegetated habitats, phytoplankton may also contribute significantly to

primary production, where the balance between water column and benthic primary production is dependent on depth, light availability, and nutrient levels, but it is unclear if total ecosystem primary production is affected by these factors (Borum and Sand-Jensen 1996).

A fundamental control on estuarine primary production is light availability, which is affected by bathymetry for benthic primary producers, but is also a function of other spatiotemporally dynamic variables (e.g., nutrient

availability, sediment type). Models of light attenuation consider the role of suspended sediment, phytoplankton, and colored dissolved organic matter (CDOM) concentrations in the water column, either through empirical formulations (Xu et al. 2005) or detailed models of scattering and absorption properties (Gallegos et al. 1990).





Suspended-sediment concentrations are controlled by processes that vary on a variety of time scales (minutes, weeks, months), including bed composition, bed shear stress, resuspension, and advective inputs of sediment from external sources. In contrast, phytoplankton concentrations are a function of light and water column nutrients, while CDOM is driven by the input of terrestrial material through freshwater loading. The relative contributions of these

constituents to light attenuation is dependent on local conditions, can vary spatially and seasonally based on external forcings (wind-induced resuspension, external inputs), and thus requires high-frequency measurements over space and time to measure all aspects of variability.

A wealth of literature describes the relationship between light and photosynthesis for marine photoautotrophs, including for SAV the role of self-shading, overall water-column conditions, and light attenuation (Sand-Jensen et

al. 2006, Kemp et al. 2004, Duarte 1991). Light-photosynthesis interactions for SAV and phytoplankton can differ substantially, given that phytoplankton are vulnerable to water-column structure and mixing while SAV are rooted to sediments, SAV are vulnerable to epiphytic fouling under nutrient-enriched conditions (Neckles et al. 1993), and SAV can engineer their own light environment through attenuation of flow velocity, wave energy, and therefore bed shear stress and resuspension (Hansen and Reidenbach 2013). Multiple studies have revealed self-reinforcing

feedbacks within SAV beds, where SAV shoots and roots stabilize the sediment bed to reduce sediment resuspension, increasing the local net deposition of water-column particulates and improving local light conditions (e.g., Gurbisz et al. 2017). These feedbacks are complex, however, and depend on bed size, aboveground biomass, and other factors (e.g., Adams et al. 2016). Consequently, healthy SAV beds are likely to generate high rates of primary production relative to adjacent unvegetated areas, but each habitat may respond differently to tidal, diurnal,

and event-scale variations in physical forcing.

The spatiotemporal variability of light attenuation and primary production in back-barrier estuaries over seasonal timescales is not well-constrained due to large gradients in benthic habitats, nutrient and carbon concentrations, and circulation. Nonetheless, numerous net ecosystem metabolism (NEM) estimates have been made with limited spatiotemporal information due to the inherent difficulty of continuous measurements at multiple locations.  For

example, Caffrey (2004) synthesized NEM across numerous estuaries, however some estuaries were represented by 1-2 sampling locations within the system. Conversely, Howarth et al. (2014) quantified NEM at multiple locations in a small, eutrophic estuary, but sampling was limited to ~ 100 d over 7 years, thereby adding uncertainty to annual


rates. Indeed, Staehr et al. (2012) suggest that undersampling and variability are both large sources of uncertainty in NEM estimates that are not well-understood.

We hypothesize that the type of benthic habitat and biophysical environment control spatial variations in primary production via their influence on light availability via local attenuation of light by resuspended particles and water-

column phytoplankton biomass. Consequently, the quantification of spatial variability in ecosystem metabolism and subsequent 'up-scaling' to system-wide rates is influenced by sampling frequency. Therefore, the objective of this study is to quantify sub-hourly variations in light attenuation, water-column properties, and net ecosystem metabolism and examine the relationship among these properties across habitats in a back-barrier estuary, Chincoteague Bay (Maryland/Virginia, USA) using a year-long deployment of high-frequency sensors. We first

describe the observational campaign and analytical methods, followed by analysis of the light attenuation and associated forcing mechanisms. We then quantify gross primary production, respiration, and net ecosystem metabolism, which have not been studied comprehensively in this estuary, and relate it with the variability of light attenuation across different habitats. Given the large spatial variability in bed sediment type, bathymetry, and dominant vegetation, we aim to quantify how temporal variations in light-attenuating substances, wave dynamics,

dissolved oxygen, and metabolic rates are linked across these spatially distinct environments. Our conclusions highlight the importance of quantifying spatiotemporal variability in these processes, which indicate feedbacks between physical and ecological processes in estuarine environments that should be considered when evaluating future ecosystem response.

## 2 Methods

### 2.1 Site description

Chincoteague Bay, a back-barrier estuary on the Maryland/Virginia Atlantic coast (Fig. 1), spans 60 km from Ocean City Inlet at the north to Chincoteague Inlet in the south. A relatively undeveloped barrier island separates the estuary from the Atlantic Ocean. The mean depth is 1.6 m, with depths exceeding 5 m in the inlets. The central basin

depths are approximately 3 m, and the eastern, back-barrier side of the bay is characterized by shallower vegetated shoals; the western side is deeper with no shoals (Fig. 1).



The coastal ocean tide range approaches 1 m, but is attenuated to less than 10 cm in the center of the bay, where water levels are dominated by wind setup and remote offshore forcing (Pritchard, 1960). River and watershed constituent inputs are minimal with the highest discharge and lowest salinities near Newport Bay in the northwest corner of the bay. Atmospheric forcing is characterized by episodic frontal passages in winter with strong northeast

winds; summer and fall exhibit gentler southwest winds. Waves within the bay are predominantly locally generated, with substantial dependence on wind direction and fetch due to the alignment of the estuary along a southwest to northeast axis.

Wazniak et al. (2007) synthesized water quality metrics for Chincoteague Bay, indicating high spatial variability in chlorophyll-a, dissolved oxygen, and nutrient concentrations. Sites in the northern portion of the bay had generally

poorer water quality, ostensibly due to higher nutrient loading and poorer flushing; Newport Bay had the lowest water quality index. Fertig et al. (2013) identified terrestrial sources of nutrients to central Chincoteague Bay, though the precise source (anthropogenic vs. naturally occurring) could not be determined.

### 2.2 Time-series of turbidity, chlorophyll-a, fDOM, dissolved oxygen, and light attenuation

Instrumentation was deployed from 10 August 2014 to 12 July 2015 (Suttles et al., 2017) at four locations in

Chincoteague Bay (Fig. 1; Table 1). Instruments were recovered, downloaded, and serviced three times during that period (October 2014, January 2015, and April 2015). Beginning in the southern portion of the estuary, site CB03 is within a seagrass meadow (primarily *Zostera marina*) on the eastern edge of the southern basin, at 1 m depth Moving northward, site CB06 is within the main channel north of the nominal boundary between the northern and southern basins, with a mud-dominated bed and no vegetation, and a depth of 3 m. Site CB10 is within a seagrass

meadow on the eastern side of the northern basin, at 1 m depth. Lastly, site CB11 is in the central basin of Newport Bay within the northwest portion of the estuary, at a depth of 2.5 m. A meteorological station was deployed at site CBWS, approximately 3.8 m above the water surface at Public Landing, Maryland on the central, western shore of Chincoteague Bay. Seagrass presence/absence was confirmed visually during instrument deployments; SAV bed area was obtained from the Virginia Institute of Marine Science Submersed Aquatic Vegetation Monitoring

Program, which mapped SAV area from aerial imagery acquired annually in 2014 and 2015 (web.vims.edu/bio/sav/; Orth et al., 2016).





The shallow-water platform described by Ganju et al. (2014) was deployed at sites CB03 and CB10, within sandy

patches of the seagrass meadows. The platform was designed to measure hydrodynamic, biogeochemical, and light

parameters in the bottom half of a 1 m water column, and consists of an RBR D|Wave recorder (+/- 0.05%

accuracy), a pair of WetLabs ECO-PARSB self-wiping photosynthetically active radiation (PAR, 400-700 nm; +/-

5% accuracy) sensors; a YSI EXO2 multi-parameter sonde measuring temperature, salinity, turbidity, dissolved

oxygen, chlorophyll-a fluorescence, fluorescing dissolved organic matter (fDOM, a proxy for CDOM), pH, and

depth (parameter-dependent accuracy available at https://www.ysi.com/EXO2); and a Nortek Aquadopp ADCP

measuring water velocity profiles (2 MHz standard at CB10 and 1 MHz high resolution at CB03; +/- 1% accuracy).

All instruments except for the upper PAR sensor were mounted at 0.15 meters above the bed (mab) on a weighted

fiberglass grate approximately 1 m x 0.5 m. The lower PAR sensor was recessed inside a PVC tube protruding from

the bottom of the frame. The upper PAR sensor was mounted at 0.45 mab; the upper and lower sensors provide an

estimate of light attenuation $K_{dPAR}$ over the PAR spectrum (400-700 nm), calculated as:

$$K_{dPAR} = -\frac{1}{dz}\ln(PAR_{lower}/PAR_{upper}) \tag{1}$$

where $dz$ is the distance between the two PAR sensors (0.3 m in this case). Light attenuation was calculated only

between the hours of 1030 and 1530, when the angle of the sun relative to the deployment location was closest to 0

degrees. All sensors sampled at 15 min intervals, except for the wave recorders, which burst-sampled at 6 Hz every

3 min. Temperature, turbidity, and inner filter effects (IFE) have been shown to alter fDOM measurements (Baker,

2005; Downing et al., 2012). We corrected fDOM measurements to account for temperature, turbidity, and IFE,

according to Downing et al. (2012). Measurements of fDOM at turbidities > 50 NTU were removed due to

interference with the fluorescence signal. Chlorophyll-a concentration was calculated from sensor-based

fluorescence measurements by regressing fluorescence against discrete measurements of chlorophyll-a made on four

dates (August and October 2014, January and April 2015) at all stations (Fig. S1). In short, water was collected in

the field at the time of sensor sampling, and filtered through 0.7 μm GF/F filters, which were wrapped in foil, and

frozen until laboratory analysis using standard methods (EPA Method 445.0). Non-photochemical quenching (NPQ)

was accounted for by removing fluorescence measurements during periods of daylight (upper PAR sensor> 150

W/m², or ~40% of the record), and interpolating to fill gaps. Platforms at sites CB06 and CB11 were identical to

platforms at CB03 and CB10 except for the omission of PAR sensors. Light attenuation at those sites was estimated



using the model of Gallegos et al. (2011), discussed below; this enabled comparison between vegetated and unvegetated sites using the same fundamental variables to compute light attenuation.

We investigated the frequency response of constituents using spectral density estimates (Welch's overlapped segment averaging estimator) and their uncertainties (Bendat and Piersol 1986) for dissolved oxygen, turbidity,

chlorophyll-a, and fDOM. Uncertainty bounds on the spectra were used to test if differences in spectral density between sites were significant, where non-overlapping uncertainty envelops suggest significant differences. We also performed wavelet coherence analyses (Grinsted et al. 2004) between site CB03 (shoal) and CB06 (channel) for each constituent; wavelet coherence indicates co-variability of two signals at varying frequencies through time, and can identify mechanistic linkages between processes that have complex temporal coupling. The analysis was limited

to these two sites due to the more complete data coverage. Statistics of the constituent time-series (mean, maxima, and minima) were computed with the original 15 min data, and with variable sampling intervals (1 h, 2 h, 24 h) to investigate the influence of sampling resolution on spatial gradients in biogeochemical variables.

Further details on collection protocols and access to the time-series data are reported by Suttles et al. (2017). The hydrodynamic results of this field campaign have been described in detail by prior studies (Ganju et al. 2016;

Beudin et al. 2017; Nowacki and Ganju 2018); for the purposes of this paper we focus on the spatiotemporal variability of constituent concentrations and light attenuation.

**2.3 Estimation of light attenuation contributions**

We estimated light attenuation (for periods with missing PAR data or at sites with no PAR data) and relative contributions from turbidity, chlorophyll-a, and CDOM using the method of Gallegos et al. (2011). This formulation

computes spectral attenuation in terms of suspended and dissolved constituents including the effects of water, CDOM, phytoplankton, and non-algal particulates (NAP, e.g., detritus, minerals, bacteria). We include absorption by four components: (1) absorption by water was assumed to follow the spectral characteristics of pure water; (2) CDOM absorption was taken proportional to fDOM concentration, with a negative spectral slope (Bricaud et al. 1981) set to $s_g$=0.02 nm$^{-1}$ (Oestreich et al., 2016); (3) phytoplankton absorption was proportional to chlorophyll-a

concentration and with the spectrum shape normalized by the absorption peak at 675 nm (initial value for peak absorption was taken as $a_{\psi,675}$=0.0235 m$^2$ (mg chl $a$)$^{-1}$, within the range provided by Bricaud et al. 1995); and (4) non-algal absorption was taken as proportional to the suspended-sediment concentration with a spectral shape



(Bowers and Binding, 2006) that included a baseline of $c_{x1}$=0.0024 m$^2$ g$^{-1}$ (Biber et al. 2008), an absorption cross-section of $c_{x2}$=0.04 m$^2$ g$^{-1}$ (Bowers and Binding 2006), and a spectral slope of $s_x$=0.009 nm$^{-1}$ (Boss et al. 2001). The backscattering ratio of water was set at 0.5, while CDOM is considered non-scattering (Mobley and Stramski 1997), and the particulate effective backscattering ratio $b_{bx}$ was initially set at 0.017.

Given the high variability in turbidity and suspended-sediment concentrations due to wind-wave resuspension, we varied the value of $b_{bx}$ as a function of turbidity to achieve the best agreement between the model and observations. At turbidity below 50 NTU, $b_{bx}$ is constant at 0.017, and then linearly declines to 0.0024 at turbidities above 220 NTU. This modification implies that backscattering by mineral particles dominates at low turbidities, while large organic aggregates dominate at the highest turbidities (Gallegos et al. 2011), and has the effect of enhancing

attenuation by particulate-induced absorption at the highest turbidities. The lowest values of backscattering to scattering ratio $b_{bx}$ in the literature range from 0.005 (Snyder et al. 2008) to about 0.002 (Chang et al. 2004). The chosen value of 0.0024 was obtained from Loisel et al. (2007). Morel and Bricaud (1981) described the $b_{bx}$ ratio as decreasing with increasing absorption, which would be consistent with high turbidity situations. McKee et al. (2009) described a decrease of the $b_{bx}$ ratio with increasing concentrations in a mineral-rich environment. The maximum

sediment concentrations (15 mg/l) in that study were higher than previously mentioned studies, but smaller than concentrations observed in this environment (Nowacki and Ganju 2018). Determining the appropriate backscattering ratio at high turbidities is still poorly constrained and the minimum value used in this study might even be an overestimation. The calibration of the light model at shoal sites (CB03 and CB10) and subsequent application at channel sites assumes that the vertical variability in the bottom 0.3 m of the water column is similar among these

sites.

## 2.4 Net ecosystem metabolism

The basic concept and method for computing community production and respiration (and ecosystem metabolism) was developed by Odum and Hoskin (1958) and, with numerous modifications, has been used since for estimating these rate processes in streams, rivers, lakes, estuaries and the open ocean (Caffrey 2004; Howarth et al. 2014). The

technique is based on quantifying increases in oxygen concentrations during daylight hours and declines during nighttime hours as ecosystem rates of net primary production and respiration, respectively. The sum of these two processes over 24 h, after correcting for air-sea exchange, provides an estimate of net ecosystem metabolism. We



utilized continuous oxygen concentration measurements at four locations in Chincoteague Bay (CB03, CB10, CB06, CB11) to compare differences in net ecosystem metabolism across sites across habitats (phytoplankton versus SAV dominated, channel versus shoal). We computed daily estimates of gross primary production ($P_g$), ecosystem respiration ($R_t$), and net ecosystem metabolism ($P_{n} = P_g - R_t$) using the approach of Beck et al. (2015), which utilizes

weighted regression to remove tidal effects on dissolved oxygen time-series. The changes in dissolved oxygen concentrations used to compute metabolic rates were corrected for air-water gas exchange using the equation

$$D = K_a (C_s - C)$$

where $D$ is the rate of air-water oxygen exchange (mg $O_2$ L$^{-1}$ h$^{-1}$), $K_a$ is the volumetric aeration coefficient (h$^{-1}$), and $C_s$ and $C$ are the oxygen saturation concentration and observed oxygen concentration (mg $O_2$ L$^{-1}$), respectively. $K_a$

was computed as a function of wind speed measured at a weather station installed at a dock (near CB10; Suttles et al. 2017). Details of the air-water gas calculation are incorporated into the R package WtRegDO (Beck et al. 2015) and described in detail in Thebault et al. (2008). The calculations utilized salinity, temperature, and dissolved oxygen times-series from the sensors at each platform, and atmospheric pressure and air temperature data from a nearby buoy (OCIM2 - 8570283 at the Ocean City Inlet, Maryland).

The oxygen data used to make metabolic computations were obtained from sensors deployed near-bottom in relatively shallow waters. Our metabolic computations assume that the water-column is well-mixed, which is necessary for the air-water flux correction to be valid and for the oxygen time-series to be representative of the combined water-column and sediments (e.g., Murrell et al. 2018). We tested the validity of this assumption at our two deeper sites (which were 2-3 m deep) using monthly vertical profile data for dissolved oxygen, temperature, and

salinity from the Maryland Department of Natural Resources from 1999-2014. These data indicate instantaneous vertical dissolved oxygen differences of >1 mg/L on occasion over the 15 y record, generally during the productive summer months. The long-term mean vertical oxygen difference, however, was less than 0.5 mg/L during September to May, but between 0.5 and 0.9 mg/L during June-August, indicating that although we did not measure vertical gradients during our study, they do occasionally develop.


## 3 Results

### 3.1 Time-series of turbidity, chlorophyll-a, fDOM, dissolved oxygen, and light attenuation

Turbidity ranged from near zero to a maximum of over 400 NTU at site CB06 during a winter storm (December 2014) that induced waves exceeding 0.7 m (Figs. 2-5). The highest turbidities were observed at CB06, while sites

CB03, CB10, and CB11 had similar statistical distributions of turbidity (Table 3). Turbidity at shoal sites was highest in the winter, between November-April, while turbidity at site CB06 did not display a similar seasonal signal. In general, tidal resuspension appeared to be minimal although tidal advection after large wind-wave resuspension events was observed (Figs. 2-5). Spectrally, most of the energy in the turbidity signal was found at subtidal frequencies (i.e., > 1 day), with little correspondence between sites at tidal frequencies (Fig. 6).

Chlorophyll-a concentrations peaked just below 50 µg/L at site CB11, and below 30 µg/L at all other sites (Figs. 2-5). Chlorophyll-a concentrations were comparable across sites CB03, CB10, and CB06, but on average twice as high at CB11 (Table 1). The largest concentrations were observed in winter during resuspension events and during a broad spring bloom during March-April 2015. All sites showed a large decrease in chlorophyll-a concentration during a period of ice formation in late February 2015, when advection and resuspension were largely halted in the

estuary. Despite removing effects of NPQ, an attenuated diel signal remains in the spectra that may be a partial artifact or consistent with day-night variations in PAR (Fig. 6). The diurnal signal was comparable across all stations and was stronger than the tidal signal (Fig. 6).

Concentrations of fluorescing dissolved organic matter (fDOM) varied between 0 and 70 QSU, with the highest values consistently observed at CB11 in Newport Bay (Figs. 2-5). Concentrations were similar between the other

sites, with lowest fDOM in the winter, possibly due to reduced biological activity. Periodic large decreases in fDOM coincided with increases in turbidity due to wind-wave resuspension; despite correcting the fDOM time-series for turbidity interference, the fluorescence measurement was likely attenuated beyond correction. Most of the energy in the fDOM signal was at subtidal frequencies (Fig. 6), there was a distinct peak at the $M_2$ tidal frequency (12.42 h) corresponding to advection of fresher, fDOM-elevated water on ebb tides.

Dissolved oxygen percent saturation ranged from a minimum of 19% at site CB11 in the summer, to a maximum of over 200% at site CB10 (Fig. 7), with maximum diel variability in the summer. The channel sites (CB06 and CB11)



showed substantially attenuated diel variations as compared to the shoal sites (CB03 and CB10). Diel fluctuations in the winter were typically less than the subtidal changes. Spectral analysis clearly shows the dominance of 24 h diel fluctuations at all sites (Fig. 6), with higher energy at the vegetated shoal sites (CB03, CB10).

Direct measurements of $K_{dPAR}$ at sites CB03 and CB10 were partially confounded by instrument fouling and

malfunction of one or both sensors on each platform. Nonetheless, we successfully captured a wide range of conditions, with peak light attenuation of approximately 10 m$^{-1}$ occurring at sites CB03 and CB10 in the winter during a sediment resuspension event (Figs. 2-5). Median $K_{dPAR}$ was approximately 1 m$^{-1}$ at both shoal sites, but event-driven magnitudes exceeded 2 m$^{-1}$ multiple times during the deployment. The $K_{dPAR}$ data were used to calibrate the light model (Fig. S2), which we implemented to reconstruct missing data at sites CB03 and CB10, and

to estimate light attenuation at sites CB06 and CB11 where constituent concentrations were measured but no light data were collected. The full time-series of estimated light attenuation at the four sites demonstrates the strong seasonal variability of light attenuation at all sites (Figs. 2-5), with peak attenuation occurring during winter storms. Wave-induced sediment resuspension and advection were responsible for increased turbidity, which accounted for approximately 40% of the light attenuation at sites CB03, CB10, and CB11, and 61% at site CB06. Light attenuation

at site CB11, with its proximity to freshwater and nutrient sources, was highest overall and more highly influenced by chlorophyll-a and fDOM than at other sites. Median $K_{dPAR}$ was highest at the two unvegetated channel sites, and lowest at the two vegetated shoal sites (Table 2). Turbidity, the strongest driver of $K_{dPAR}$, was generally lower during the warm season at the vegetated shoal sites and the turbidity generated for a given wind-wave induced bed shear stress was significantly reduced in summer at the vegetated sites, while there was no significant seasonal

variation in the turbidity-shear stress relationship at unvegetated channel sites (Fig. 8).

Wavelet coherence of dissolved oxygen between site CB03 and site CB06 was maximized at diel timescales (Fig. 9), corresponding to co-varying oscillations due to daytime production and night-time respiration. Coherence was also high at lower frequencies, especially during winter when oscillations at both sites were small due to reduced production and respiration. Turbidity was coherent between channel and shoal sites primarily at subtidal timescales,

corresponding to episodic multi-day resuspension events and generally high turbidity during most of the winter. Phase lag was minimal during times of high coherence for both of these parameters. Coherence for both chlorophyll-a and fDOM was minimal throughout the year, though the former demonstrated sporadic coherence at diel and





multi-day timescales. The December-April period of generally high chlorophyll-a was similar between these sites and manifested as increased coherence at ~512-1024 h (21-42 d).

Statistics of the constituent time-series were computed with temporal sampling intervals of 15 min (original sampling interval), 1 h, 2 h, and 24 h (Figure 10; Table 3). While mean values were minimally altered, minima and

maxima were significantly dampened for dissolved oxygen and turbidity. At site CB11, the original sampling interval captured a hypoxic event in September 2014 with a minimum value of 19% dissolved oxygen, while daily sampling yielded a minimum of 55%, above the hypoxic threshold. Similarly, at site CB10 supersaturation led to a maximum of over 200% dissolved oxygen, while daily sampling yielded a maximum of 152%. Maximum values of turbidity were similarly attenuated by daily sampling (Table 3). The effect of these sampling intervals is discussed

later in the manuscript.

**3.2 Net ecosystem metabolism**

Estimates of ecosystem metabolism displayed strong seasonal variability, with elevated rates of $P_g$ and $R_t$ during warm months across all stations (Figs. 11, 12; Table S1). High rates of $P_g$ and $R_t$ persisted between May and October, when temperature and PAR peaked seasonally, and were consistently lower during November to April

(Figs. 11, 12), and $R_t$ was exponentially related to temperature across sites (although less so at CB06). Metabolic rates were clearly higher at vegetated shoal sites (CB03, CB10), consistent with the strong diurnal signal in oxygen at these sites (Figs. 6, 7) and the presence of seagrass (Table 1). Temperature was strongly associated with rates of respiration across all sites, and respiration reached peak values under conditions of high temperatures and high rates of $P_g$ (Fig. 12). Gross primary production and respiration were largely balanced across all sites, but instances of net

autotrophy ($P_g > R_t$) occurred nearly 70% of the time at the vegetated sites, and were also persistent at CB06 (Fig. 13). $P_g$ and $R_t$ were more balanced at CB11, but we did not have enough data to capture the complete seasonal cycle at this site (Fig. 11).

The relationship between rates of $P_g$ and light availability was site specific. At the vegetated sites, CB03 and CB10, $P_g$ and $K_{dPAR}$ were significantly negatively correlated, with high rates of $P_g$ occurring during the periods of lowest

$K_{dPAR}$ and highest surface PAR (Fig. 14). At CB06 and CB11, variations in $P_g$ and $K_{dPAR}$ were not significantly correlated and $K_{dPAR}$ was higher and $P_g$ was lower at these sites than at CB03 and CB10 (Tables 1, 2; Figs. 13, 14). In summary, the highest metabolic rates we measured occurred during warm periods in vegetated shoals, where





wave attenuation by seagrass reduced turbidity and $K_{dPAR}$.

## 4 Discussion

Net ecosystem metabolism and other water-quality parameters are typically measured over limited spatial (e.g.

single point) and temporal (e.g. days-to-weeks) scales. An analysis of a comprehensive suite of high-frequency

biological and physical measurements in Chincoteague Bay over an annual cycle revealed the primary drivers of

light attenuation, the role of light attenuation in driving variations in gross primary production, the primary

timescales of biogeochemical variability, and the effect of habitat type (i.e., vegetated versus un-vegetated; nutrient

enrichment) on oxygen variability and net ecosystem metabolism. Turbidity dominated light attenuation variability

and varied considerably at 1-7 d time scales, consistent with the frequency of storm passage. Storm-associated wind

waves were the specific driver of resuspension and turbidity, and reduction of bed shear stress and turbidity in SAV-

dominated shoal environments during summer increased light availability in these habitats. As a consequence, $P_g$

was substantially higher in SAV-dominated shoals compared with adjacent plankton-dominated sites and $P_g$ was

negatively correlated with $K_{dPAR}$ in the shoals, highlighting the role of short-term variability in light availability

driving $P_g$. High rates of $P_g$ and $R_t$ in shoal environments led to much higher diurnal and seasonal-scale variability in

dissolved oxygen in these habitats.

### 4.1 Spatiotemporal variability of constituents

#### 4.1.1 Interpretation of spectral signals

Spectral analysis of the constituent time-series elucidates significant differences between channel and shoal sites, as

well as longitudinally within the estuary. With regards to dissolved oxygen, the peak in diurnal energy at shoal sites

is markedly stronger relative to channel sites with no overlap in uncertainty bounds between channel and shoal sites,

indicating higher local production/respiration due to SAV presence. Between sites CB03 and CB06, the diurnal peak

in spectral density is reduced by 94%, and reduced by 83% between site CB10 and CB11. The peak in spectral

density was 30% higher at CB03 than CB10, perhaps due to higher SAV biomass at that site. This indicates the

dominant role of seagrass in controlling oxygen concentrations locally, and the relatively weaker oxygen dynamics



at the channelized, unvegetated sites. From a sampling perspective, this also suggests that interpreting oxygen

dynamics with limited spatial resolution is confounded by heterogeneous benthic coverage.

The highest spectral densities for turbidity were observed at site CB06, across all frequencies. At the lowest

frequencies, corresponding to multi-day storms, there was no overlap in the uncertainty bounds. Nowacki and Ganju

(2018) showed that sediment fluxes at this site were an order of magnitude larger than the other three sites, due to a

strong response to wind events, which are typically multi-day events. The central location of the site suggests that

local sediment transport would respond consistently to the seasonally variable winds, which tend to act along the

axis of the estuary. Resuspension over shoals on either end of the estuary, and subsequent advection cause the

spatial integration of resuspension processes throughout the estuary at this main channel site.

Spectral density peaks of chlorophyll-a were similar between sites with significant overlap across frequencies, but

site CB11 demonstrated the highest total spectral density, primarily at subtidal frequencies. The reduced exchange

with other portions of the estuary, local nutrient inputs, and longer residence times increase the potential for

phytoplankton growth at this station, which is known to be locally eutrophic (Fertig et al. 2009). This coincides with

the lowest oxygen minima across the four sites (Table 3), indicating eutrophication. Again, in regards to sampling

strategies, characterizing spatial gradients in eutrophication is confounded by the competing timescales of transport

and biogeochemical processes.

While the tidal signal in fDOM was pronounced (peak at 12.4 h), the bulk of the spectral density was in the lower

frequency band at sites CB10 and CB11, in the northern half of the estuary. This likely represents a longitudinal

gradient in freshwater input, with decreased salinity in the north where inputs are largest. The inverse correlation

between salinity and fDOM in Chincoteague Bay was identified in prior work (Oestreich et al. 2016), and highlights

the roles of tidal circulation, residence time, and freshwater input in biogeochemical and light attenuation patterns.

### 4.1.2 Channel-shoal coherence of water quality parameters

Wavelet coherence analyses demonstrate that though certain parameters are tightly coupled between channel and

shoal, other parameters are not. The coherence of dissolved oxygen at diel and longer timescales, along with nearly

no phase lag, is congruent with diel and seasonal patterns of production and respiration. The "breathing" of the

estuary (Odum and Hoskin, 1958) appears as a near-simultaneous process regardless of location, and demonstrates



the importance of local biogeochemical processes throughout the system. With regards to turbidity and sediment resuspension, the coherence at subtidal timescales indicates that channels integrate resuspension processes over the entire estuary; minimal phase lag suggests that a rapidly evolving local wave climate in response to episodic winds controls both resuspension and rapid advection through the channels (Nowacki and Ganju, 2018). Conversely,

chlorophyll-a demonstrates coherence only over the seasonal timescale, and therefore advection and other tidal processes do not appear to control dynamic exchange of phytoplankton or benthic microalgae. Lastly, the lack of coherence in fDOM shows that freshwater input is relatively minor in the southern half of the estuary, and despite a significant peak in spectral density at the tidal timescale, the advection of freshwater (and fDOM) between channel and shoal is not significant. Despite a qualitatively coherent annual signal in fDOM, the analysis cannot detect

coherence at this timescale with a single year of data. These complex relationships between parameters highlight both the intricate response to spatiotemporal variability in biophysical environments, and the necessity of high-frequency, spatially comprehensive measurements to understand estuarine ecosystem behavior.

*4.1.3 Influence of temporal sampling resolution on spatial gradients*

Though mean biogeochemical values were relatively insensitive to sampling interval, maximum values were

significantly modulated for all parameters, while minimum values for dissolved oxygen were also affected. With regards to dissolved oxygen specifically, we find that daily sampling dampens the spatial variability in maxima and minima between sites. For example, a daily sampling program would not detect hypoxic conditions at site CB11, and would only observe a 13% difference in dissolved oxygen as compared to nearby site CB10, while the actual difference was over 30%. This is relevant given the ubiquitous daily sampling programs in many estuaries, which

cover numerous locations with infrequent sampling. This result suggests that characterizing differences in water-column conditions across space requires sampling at timescales finer than 1 day, especially in highly metabolic environments where diel oscillations in dissolved oxygen are large. Continuous monitoring at multiple locations in an estuary is difficult due to fouling and instrument limitations (e.g. power and memory), especially across large systems over an entire year. However, it may be more informative to conduct shorter term, continuous monitoring

deployments at select locations rather than daily monitoring at many locations. In the case of Chincoteague Bay, characterizing the oxygen environment would be possible with seasonal, month-long deployments at four sites. This




type of data would yield both an accurate assessment of minimum and maximum values, and allow for spectral analysis to capture tidal timescale variability.

### 4.2 Spatiotemporal variability of light attenuation

Light attenuation can appear to be controlled by uncoupled biological (i.e., nutrient loading and phytoplankton

blooms) and physical (sediment resuspension) processes, but in reality the feedbacks between physics and biology are consistently present in estuaries. Wave-induced suspended-sediment resuspension is primarily responsible for light attenuation in shallow lagoons like Chincoteague Bay, and seagrass clearly modulates the magnitude and spatiotemporal variability in sediment resuspension through the attenuation of wave energy (e.g., Hansen and Reidenbach 2013). This wave-induced resuspension and turbidity occurs on the time scale of periodic wind events

observed in this system. During summer months, when vegetation densities are ostensibly the highest, the dependence (i.e. slope) of turbidity on bed shear stress decreases significantly at the vegetated shoal sites, while it is not significantly different at the unvegetated channel sites (Fig. 8). In the winter, when seagrass is largely absent, the slope increases at the shoal sites, indicating the influence of seagrass on bed stabilization in the summer. The role of seagrass in modulating their physical environment is demonstrated by this improvement in light availability when

vegetation re-establishes in the warmer months. Furthermore, the dependence of $P_g$ on $K_{dPAR}$ at the vegetated shoal sites (Fig. 14) completes the positive feedback loop between seagrass, sediment resuspension, light attenuation, and primary production.

Our measurements underscore the ability of comprehensive continuous measurements to capture multiple scales of variation in light attenuation. For example, subsampling our light attenuation measurements under fair weather

conditions (wave height < median) leads to underestimation of median $K_{dPAR}$ by as much as 27% (sites CB06 and CB10). Subsampling to periods with wave heights less than the 84th percentile leads to underestimation by 13% (site CB06). The effect of these changes can be large if sparse data are used to assess trends in biogeochemical variables, or to drive ecological models. In situations where continuous monitoring of light attenuation is difficult, light modeling using continuous measurements of turbidity, chlorophyll-a, and fDOM is a suitable proxy. With a

reasonable number of discrete light attenuation samples, it is possible to calibrate a light model and estimate seasonal changes in contributions to light attenuation using the constituent measurements. This becomes useful





when attempting to determine the relative influence of physical and biogeochemical processes on spatiotemporal

variations in light attenuation, and the potential benefits of restoration activities.

Apart from Newport Bay in the northwest corner of the estuary, Chincoteague Bay is relatively unimpacted by

external nutrient and organic material inputs, as compared to other lagoonal systems on the U.S. East Coast (e.g.

Indian River Lagoon, Phlips et al. 2002; Great South Bay, Kinney and Valiela 2011). For example, the relatively

urbanized watershed surrounding Barnegat Bay, New Jersey, contributes larger nutrient loads to the northern portion

of the estuary (Kennish et al. 2007), however the highest light attenuation is observed in the southern portion, where

sediment concentration are highest due to the availability of fine bed sediment and wind-wave resuspension (Ganju

et al. 2014). In fact, in Barnegat Bay light attenuation is lowest in the more eutrophic, poorly flushed (Defne and

Ganju 2015) northern portion due to coarser bed sediments and smaller wave heights (and less resuspension). In the

more nutrient-enriched regions of the Maryland coastal bays north of Chincoteague Bay, chlorophyll-a is also much

higher and likely makes a larger contribution to light attenuation (Boynton et al. 1996). In contrast, spatial gradients

in the light field within Chincoteague Bay are driven by geomorphology (depth), the presence of submerged aquatic

vegetation (CB03 and CB10), and the higher contribution of chlorophyll-a at a nutrient-enriched site (CB11;

Boynton et al. 1996). Thus, generalizing the light attenuation in back-barrier estuaries is hampered by these subtle,

habitat-specific controls on attenuation, but improvements in continuous monitoring will lead to an increased

understanding of these controls.

**4.3 Spatiotemporal variability in metabolism and relationship with benthic ecosystem and light attenuation**

Clear differences in the magnitude and timescales of oxygen dynamics were present across habitats in Chincoteague

Bay. Spectral density at the diurnal time-scale for dissolved oxygen was significantly higher at the two vegetated

sites, where high rates of primary production during the day and associated respiration at night led to large changes

in dissolved oxygen. Although this pattern is unsurprising given the high metabolic rates in these dense seagrass

beds, this study is one of few studies that have clearly documented this pattern inside and outside of SAV beds

within an estuary over annual timescales. Elevated metabolic rates in macrophyte dominated habitats relative to

phytoplankton dominated habitats have been documented in other systems (D'Avanzo et al. 1996), where the C:N

molar ratio of phytoplankton (C:N = 6.6) is much less than that measured for Chincoteague Bay *Z. marina* (C:N =



25.8±7.2), indicating higher carbon and oxygen metabolism for a given amount of nitrogen (e.g., Atkinson and Smith 1983). Elevated magnitudes of daily oxygen change have also been associated with elevated phytoplankton biomass in Chincoteague Bay (e.g., Boynton et al. 1996) and other systems (e.g., D'Avanzo et al. 1996). This is consistent with the fact that the site at Newport Bay (CB11), which had the highest plankton chlorophyll-a

concentrations across all sites, had a larger spectral density for dissolved oxygen at the diel frequency than at CB06. Newport Bay is known to have elevated nutrient inputs and concentrations associated with land-use in its watershed (Boynton et al. 1996), and spectral density for chlorophyll-a was highest at this location (Fig. 6), suggesting that this site is responding to eutrophication.

Seasonal variations in metabolic rate estimates were typical of temperate regions, but were variable across space. At

all stations, rates of $P_g$ and $R_t$ were highest in warmer months when incident PAR and temperature are highest in this region (e.g., Fisher et al. 2003), but rates were especially high in the later summer (August and September) at the vegetated sites. This late-summer period is typically when SAV biomass is highest and where the self-reinforcing effect of wave attenuation, reduced resuspension, and reduced $K_{dPAR}$ allow for high rates of primary production (Figs. 8, 14). Indeed, the relationship between $K_{dPAR}$ and $P_g$ was strongest at these vegetated sites (Fig. 14),

displaying the sensitivity to benthic primary production to light attenuation, which was dominated by turbidity. Elevated temperatures during this period also stimulate respiration (e.g., Marsh et al. 1986), which has been documented at the ecosystem scale in other SAV-dominated coastal systems (Howarth et al. 2014, Boynton et al. 2014). Seasonal variations in metabolic rates were comparatively lower at the unvegetated, non-eutrophic site (CB06), where low phytoplankton biomass and the absence of macrophytes leads to limited primary production and

respiration rates.

The metabolic balance (ratio) between primary production and respiration is a metric of interest in coastal aquatic ecosystems as it provides an indication of whether the system is a relative source or sink of carbon with respect to the atmosphere (Stæhr et al. 2006). Many river-dominated estuaries and estuaries flanked by extensive tidal marshes are net heterotrophic, where respiration exceeds primary production (e.g., Kemp and Testa 2011). $P_g$ and $R_t$ are often

balanced or indicate modest autotrophy in shallow lagoons where SAV is present (e.g., Ferguson and Eyre 2010, D'Avanzo et al. 1996). Autotrophy tends to persist in these habitats as submerged macrophytes like SAV generate high biomass where light availability is relatively high given low nutrient concentrations, low phytoplankton



biomass, and sediment trapping within SAV beds. In fact, modest net autotrophy prevailed during the summer season at vegetated sites but not at un-vegetated sites. The rates of $P_g$ and $R_t$ we estimated, which were typically between 200 and 400 mmol $O_2$ $m^{-2}$ $d^{-1}$, are comparable to those measured in other temperate ecosystems dominated by *Zostera marina* (Howarth et al. 2014) and net autotrophy in these low-nutrient environments is consistent with

other coastal lagoons in the region (Giordano et al. 2012, Stutes et al. 2007). While the overall metabolic balance ($P_g/R_t$) of Chincoteague Bay is difficult to assess, given large differences in metabolic balance across gradients of depth and nutrient enrichment, we made a simple calculation where mean rates of net ecosystem metabolism at CB10 were multiplied by the area of SAV in 2015 (30 $km^2$) and rates from CB06 are multiplied by the remaining area without SAV (276 $km^2$). The calculation assumes that rates measured at these stations are representative of

similar habitats across Chincoteague Bay, and does not account for the potential of benthic photosynthesis to occur in shallow regions not occupied by SAV. This approach generates an estimate of net metabolism for Chincoteague Bay of 317±461 Mmol $O_2$ $yr^{-1}$, which indicates autotrophy basin-wide, but the uncertainty is high. Given ongoing loss of SAV in this and other back-barrier systems, it is likely that the metabolic balance may shift towards heterotrophy in the future.

$P_g$ and $R_t$ were tightly coupled in the Chincoteague Bay stations during 2014-2015, suggesting that primary producers were the dominant sources of respiration in the ecosystem. At the vegetated stations, low plankton chlorophyll-a levels indicate that SAV were the dominant contributors to ecosystem metabolism, though macroalgae and epiphytic algae were also present. Alternatively, high chlorophyll-a levels at CB11 (Newport Bay) suggest that phytoplankton were dominant contributors to metabolism, as this site is known to be nutrient enriched. Alternate

sources of primary production and respiration that could drive metabolism include benthic micro- and macro-algae and heterotrophic bacteria. Previously measured sediment oxygen uptake (SOU) rates during summer in sediment incubations without light suggest a mean SOU of 56.2 (±24.7) mmol $O_2$ $m^{-2}$ $d^{-1}$ (Bailey et al. 2005) which is ~50% of the mean summer respiration rates at CB06, but only 15% of rates measured at the phytoplankton-dominated CB11. Rates of benthic photosynthesis were not available at this site, but with a mean $K_{dPAR}$ of 1.35 at CB06 and a

water-column depth of 3 m, only a small fraction of surface PAR would be expected to reach the sediments. Thus, respiration at CB06 might be evenly split between the water-column and sediments, which fits well with previous cross-system comparisons of coastal marine ecosystems (Kemp et al. 1992, Boynton et al. 2018).




**5 Conclusions**

We tested the hypothesis that spatiotemporal differences in benthic habitat would exert a strong control on biogeochemical variables and primary production in an otherwise well-mixed estuary. We found a clear linkage between SAV presence/absence, dependence of turbidity on shear stress, the ensuing light attenuation, and gross

primary production, reinforcing the positive feedback loop between seagrass presence and primary production. Vegetated shoal sites exhibited higher metabolic rates, reduced sediment resuspension, and reduced light attenuation as compared to unvegetated channel sites. Light attenuation was dominated by wind-wave induced sediment resuspension, which was minimized during summer months in vegetated shoal sites when SAV aboveground biomass was at its peak. Furthermore, we found a clear causal linkage between SAV presence, the reduction of

turbidity at a given shear stress, lowered light attenuation, and elevated gross primary production, reinforcing the positive feedback loop between seagrass presence and primary production. As a consequence, despite relatively balanced ecosystem production and respiration across all sites, high rates of primary production led to modest net autotrophy at sites vegetated by SAV. Future shifts towards phytoplankton-dominant habitat, given ongoing SAV loss, will likely shift these systems towards net heterotrophy and reduce their effectiveness as carbon sinks.

Interpreting spatial differences in biogeochemistry was strongly influenced by temporal sampling resolution, where daily sampling reduced spatial variability between sites and attenuated peak values significantly. These results demonstrate the value of high-temporal resolution measurements at multiple locations within a given estuary, due to the strong interplay between geomorphology, light attenuation, SAV dynamics, and ecosystem metabolism. The mechanisms identified here demonstrate a need to consider feedbacks between biological and physical processes in

estuaries, especially when constructing deterministic models or evaluating future ecosystem responses to eutrophication and climate change.

**Code availability**

Freely available MATLAB toolboxes, as cited in the text, were used for analyses.

**Data availability**

The time-series data are available at https://stellwagen.er.usgs.gov/chincoteague.html



**Sample availability**

N/A

**Appendices**

N/A

**Team list**

N/A

**Author contribution**

N.K.G., S.E.S, and J.M.T. designed the study, A.L.A. implemented the light model, all authors analyzed time-series data and drafted the manuscript.

**Competing interests**

The authors declare that they have no conflict of interest.

**Disclaimer**

Use of brand names is for identification purposes only and does not constitute endorsement by the U.S. Government.

**Acknowledgments**

We thank Alexis Beudin, Dan Nowacki, Sandra Brosnahan, Dave Loewensteiner, Nick Nidzieko, Ellyn Montgomery, and Pat Dickhudt for field and data assistance. Casey Hodgkins and Amanda Moore provided help with sample processing and data analysis. W. Michael Kemp provided constructive feedback on the manuscript. This study was funded by the USGS Coastal and Marine Geology Program, the Department of the Interior Hurricane Sandy Recovery program (GS2-2D). Time-series data can be accessed at the USGS Oceanographic Time-Series

Database at http://dx.doi.org/10.5066/F7DF6PBV. This is UMCES Contribution Number XXXX. Use of brand names is for descriptive purposes only and does not constitute endorsement by the U.S. Geological Survey.



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



**Figure captions**

Figure 1. Location map with bathymetry relative to mean sea level, instrument locations, and submerged aquatic vegetation coverage from the Virginia Institute of Marine Science annual SAV mapping survey (http://web.vims.edu/bio/sav/). WQ refers to water-quality sonde; PAR refers to paired upper/lower PAR sensors;

MET refers to meteorological parameters.

Figure 2. Time-series of wave height, upper and lower PAR, KdPAR, turbidity, chlorophyll-a, and fDOM at vegetated shoal site CB03.

Figure 3. Time-series of wave height, upper and lower PAR, KdPAR, turbidity, chlorophyll-a, and fDOM at vegetated shoal site CB10.

Figure 4. Time-series of wave height, modeled KdPAR, turbidity, chlorophyll-a, and fDOM at unvegetated channel site CB06.

Figure 5. Time-series of wave height, modeled KdPAR, turbidity, chlorophyll-a, and fDOM at unvegetated channel site CB11.

Figure 6. Spectral density estimates for dissolved oxygen, turbidity, chlorophyll-a, and fDOM at all four sites.

Shaded areas indicate 90% uncertainty bounds of spectral density estimates.

Figure 7. Time-series of dissolved oxygen from four sites.

Figure 8. Relationship between combined wave-current induced bed shear stress and turbidity at four sites, with repeated-median linear regressions of bin-averaged values over summer (May-September) and winter (October-April) seasons. Larger slopes indicate a stronger relationship between shear stress and turbidity. Dashed lines

indicate 95% confidence bounds on slopes; bounds overlap at channel sites indicating a lack of significant difference between seasons, while bounds do not overlap at shoal sites indicating a significant seasonal difference in the stress-turbidity relationship, likely caused by vegetation.

Figure 9. Wavelet coherence between time-series at sites CB03 (vegetated shoal) and CB06 (unvegetated channel), for four water quality parameters. Increased coherence at a given period indicates co-variability of the time-series at

that time; for example, increased coherence at ~24 h for dissolved oxygen during most of the record indicates co-





varying diel oscillations at both sites during most of the year. Direction of arrows indicates phase; arrows pointing to the right indicate that signals are in phase.

Figure 10. Mean (dots), maxima and minima (bars) for each parameter using different temporal sampling intervals (15 min, 1 h, 2 h, 24h). Spatial gradients in dissolved oxygen are most impacted by coarse temporal resolution, with

spatial differences in minima and maxima largely eliminated at resolution of 1 d.

Figure 11. Monthly estimates of gross primary production (Pg), respiration (Rt), and net ecosystem metabolism (NEM) at four sites. Measurements span August 2014 through July 2015, therefore time axis begins in August 2014 and wraps back to January 2015.

Figure 12. Relationship between respiration (Rt) and water temperature at four sites, with data coloration scaled to

gross primary production (Pg).

Figure 13. Relationship between gross primary production (Pg) and respiration (Rt) at four sites, line of 1:1 agreement shown.

Figure 14. Relationship between light attenuation $K_{dPAR}$ and gross primary production (Pg), with coloration indicating surface photosynthetically active radiation (PAR, $\square$M m-2 s-1) measured at the weather station. Linear

model was fitted to Pg as a function of log ($K_{dPAR}$). The relationship between Pg and $K_{dPAR}$ significant at vegetated shoal sites, but not significant at unvegetated channel sites.





**Tables**

Table 1. Water-column properties at the four study sites, SAV presence/absence, and metabolic rate estimates.
Temperature data are minimum and maximum values (min, max), salinity is the annual mean, chlorophyll-a is the
annual mean, and metabolic rates estimates are means ($\pm SD$) for the August-September (peak SAV growth) period.

|  | **CB03** | **CB06** | **CB10** | **CB11** |
|---|---|---|---|---|
| *Depth ($m^{-1}$)* | 1 | 3 | 1 | 2.5 |
| *Temperature (degC)* | -1.6 – 30.8 | -1.6 – 29.1 | -1.7 – 30.1 | -1.5 – 29.2 |
| *Salinity* | 26.8 | 26.7 | 26.2 | 24.5 |
| *Chlorophyll-a (µg/L)* | 5.2 | 5.6 | 5.9 | 11 |
| *SAV presence* | Y | N | Y | N |
| *$P_g$ (mmol $O_2$ $m^{-2}$ $d^{-1}$)* | 310.7 ($\pm$162.2) | 93.2 ($\pm$53.7) | 281.9 ($\pm$198.5) | 239.9 ($\pm$133.9) |
| *$R_t$ (mmol $O_2$ $m^{-2}$ $d^{-1}$)* | 304.1 ($\pm$179.7) | 92.9 ($\pm$55.3) | 267.0 ($\pm$198.2) | 252.2 ($\pm$133.4) |
| *$P_n$ (mmol $O_2$ $m^{-2}$ $d^{-1}$)* | 6.7 ($\pm$36.8) | 0.3 ($\pm$27.1) | 14.9 ($\pm$46.6) | -12.3 ($\pm$57.8) |

Table 2. Median light attenuation and relative contributions from turbidity, chlorophyll-a, and fDOM. Remainder of
light attenuation contribution is from water (not shown), and equivalent between sites.

|  | **CB03** | **CB06** | **CB10** | **CB11** |
|---|---|---|---|---|
| *$K_{dPAR}$ ($m^{-1}$)* | 1.00 | 1.35 | 1.19 | 1.67 |
| *$K_{dPAR}$ (turbidity)* | 0.44 (44%) | 0.82 (61%) | 0.53 (45%) | 0.66 (40%) |
| *$K_{dPAR}$ (chl-a)* | 0.11 (11%) | 0.10 (7%) | 0.11 (10%) | 0.37 (22%) |
| *$K_{dPAR}$ (fDOM)* | 0.17 (17%) | 0.15 (11%) | 0.24 (20%) | 0.31 (19%) |



Table 3. Mean, minima, and maxima for four water-quality parameters at four sites with variable temporal sampling resolution.

| Site | Resolution | Dissolved oxygen (%) | | | Turbidity (NTU) | | | Chlorophyll-a (ug/L) | | | fDOM (QSU) | | |
|------|-----------|------|-----|-----|------|-----|-----|------|-----|-----|------|-----|-----|
| | | mean | min | max | mean | min | max | mean | min | max | mean | min | max |
| CB03 | 15 min | 104.4 | 40.9 | 182.8 | 14.9 | 0.0 | 427.4 | 5.2 | 0.2 | 29.9 | 12.6 | 0.0 | 30.4 |
| | 1 h | 104.4 | 41.2 | 182.3 | 14.9 | 0.0 | 396.9 | 5.2 | 0.2 | 29.7 | 12.6 | 0.0 | 29.7 |
| | 2 h | 104.4 | 44.4 | 180.1 | 14.9 | 0.0 | 396.9 | 5.2 | 0.2 | 29.0 | 12.6 | 3.4 | 29.0 |
| | 24 h | 106.5 | 59.1 | 149.2 | 16.7 | 0.0 | 242.7 | 5.5 | 0.2 | 27.6 | 12.5 | 3.7 | 25.8 |
| CB06 | 15 min | 103.0 | 70.2 | 149.4 | 27.1 | 0.0 | 546.8 | 5.6 | 0.0 | 30.0 | 6.7 | 0.0 | 18.6 |
| | 1 h | 103.0 | 75.0 | 149.0 | 27.0 | 0.0 | 546.8 | 5.6 | 0.0 | 29.7 | 6.7 | 0.0 | 18.2 |
| | 2 h | 103.0 | 77.9 | 149.0 | 27.2 | 0.0 | 525.7 | 5.6 | 0.1 | 29.1 | 6.7 | 0.0 | 17.5 |
| | 24 h | 104.2 | 92.8 | 141.5 | 28.7 | 0.0 | 356.7 | 5.2 | 0.2 | 25.4 | 6.7 | 0.0 | 17.0 |
| CB10 | 15 min | 108.1 | 53.4 | 214.5 | 13.6 | 0.0 | 375.0 | 5.9 | 0.2 | 39.5 | 16.7 | 3.4 | 39.7 |
| | 1 h | 108.1 | 55.3 | 212.0 | 13.6 | 0.0 | 308.8 | 5.9 | 0.3 | 35.9 | 16.7 | 3.4 | 39.7 |
| | 2 h | 108.1 | 56.3 | 212.0 | 13.6 | 0.0 | 308.8 | 5.8 | 0.3 | 35.8 | 16.7 | 3.4 | 39.7 |
| | 24 h | 105.1 | 68.7 | 151.9 | 15.7 | 0.0 | 274.5 | 5.0 | 0.3 | 35.8 | 16.5 | 4.2 | 37.7 |
| CB11 | 15 min | 99.3 | 19.2 | 160.3 | 12.4 | 0.0 | 275.2 | 11.0 | 0.5 | 49.6 | 26.0 | 5.4 | 56.8 |
| | 1 h | 99.3 | 19.4 | 160.3 | 12.4 | 0.0 | 255.5 | 11.1 | 0.5 | 49.0 | 26.0 | 5.5 | 56.8 |
| | 2 h | 99.3 | 20.9 | 160.3 | 12.4 | 0.0 | 255.5 | 11.1 | 0.5 | 40.9 | 26.1 | 5.5 | 52.3 |
| | 24 h | 104.7 | 55.3 | 143.9 | 14.2 | 0.0 | 174.3 | 10.8 | 0.7 | 36.8 | 25.6 | 8.1 | 46.9 |





**Figures**

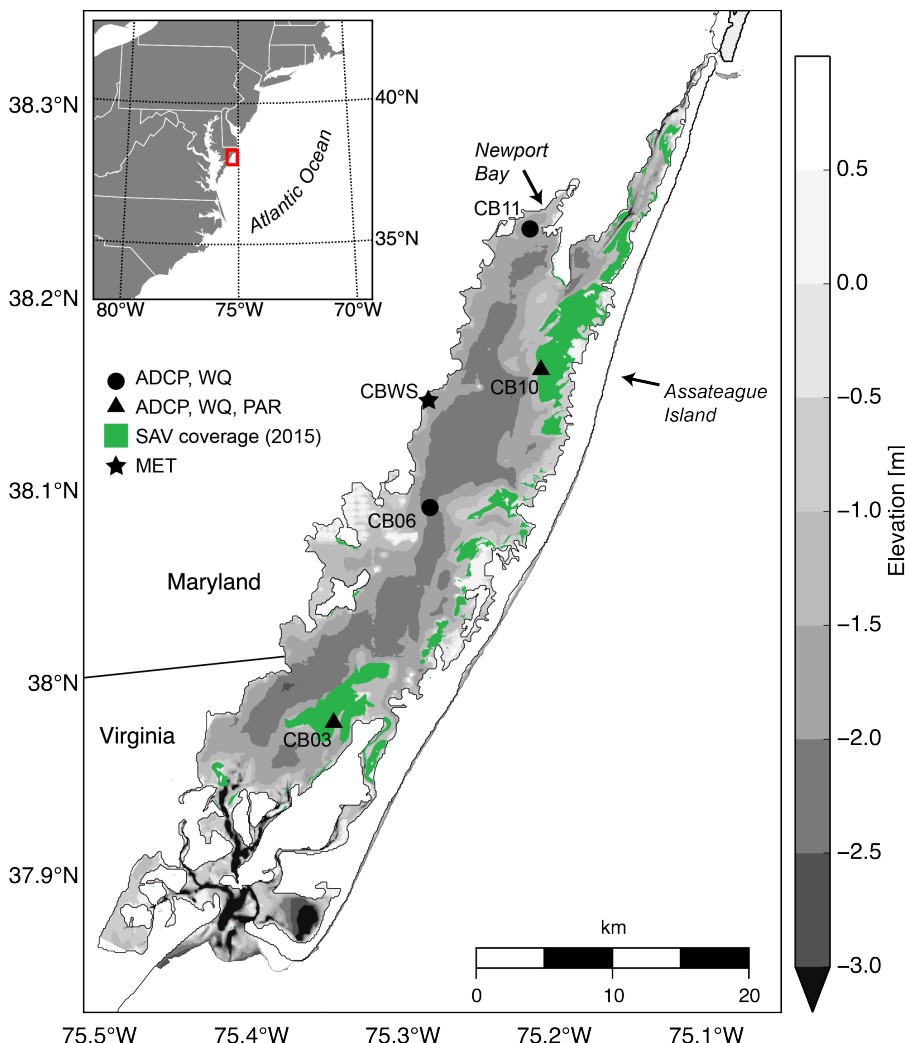

Figure 1. Location map with bathymetry relative to mean sea level, instrument locations, and submerged aquatic

vegetation coverage from the Virginia Institute of Marine Science annual SAV mapping survey

5    (http://web.vims.edu/bio/sav/). WQ refers to water-quality sonde; PAR refers to paired upper/lower PAR sensors;

MET refers to meteorological parameters.



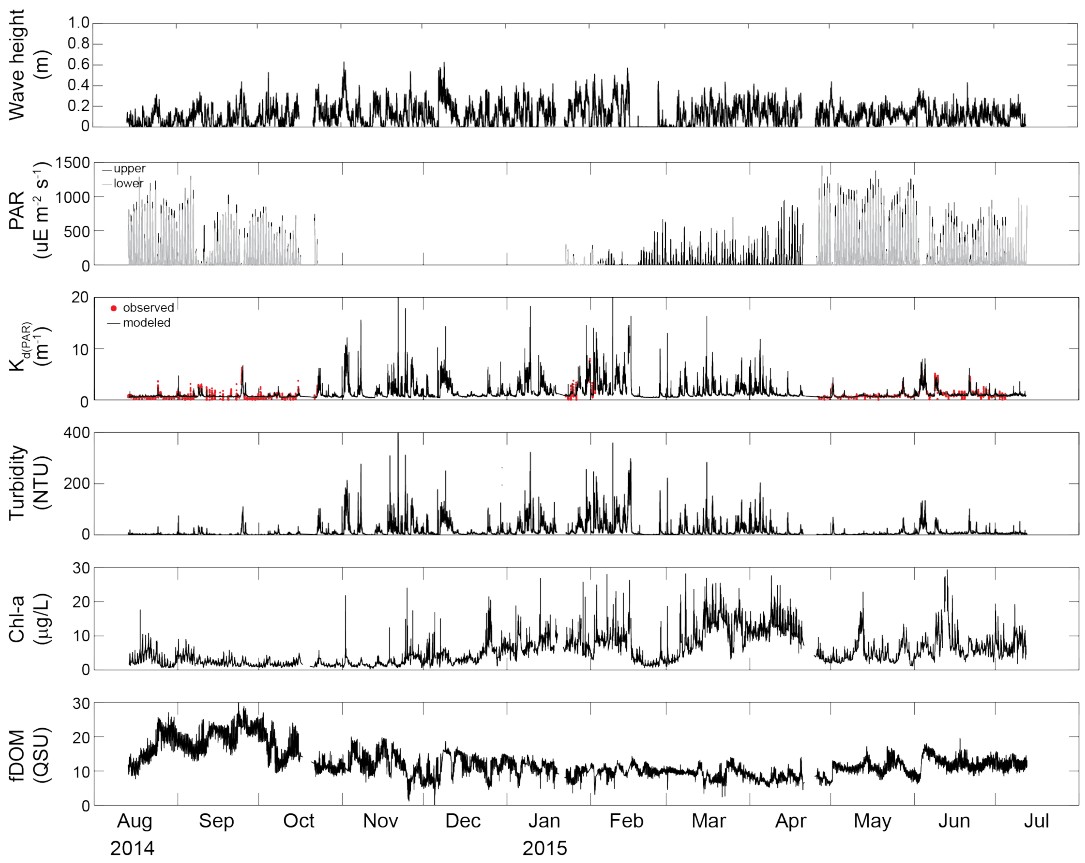

Figure 2. Time-series of wave height, upper and lower PAR, $K_{dPAR}$, turbidity, chlorophyll-a, and fDOM at vegetated shoal site CB03.



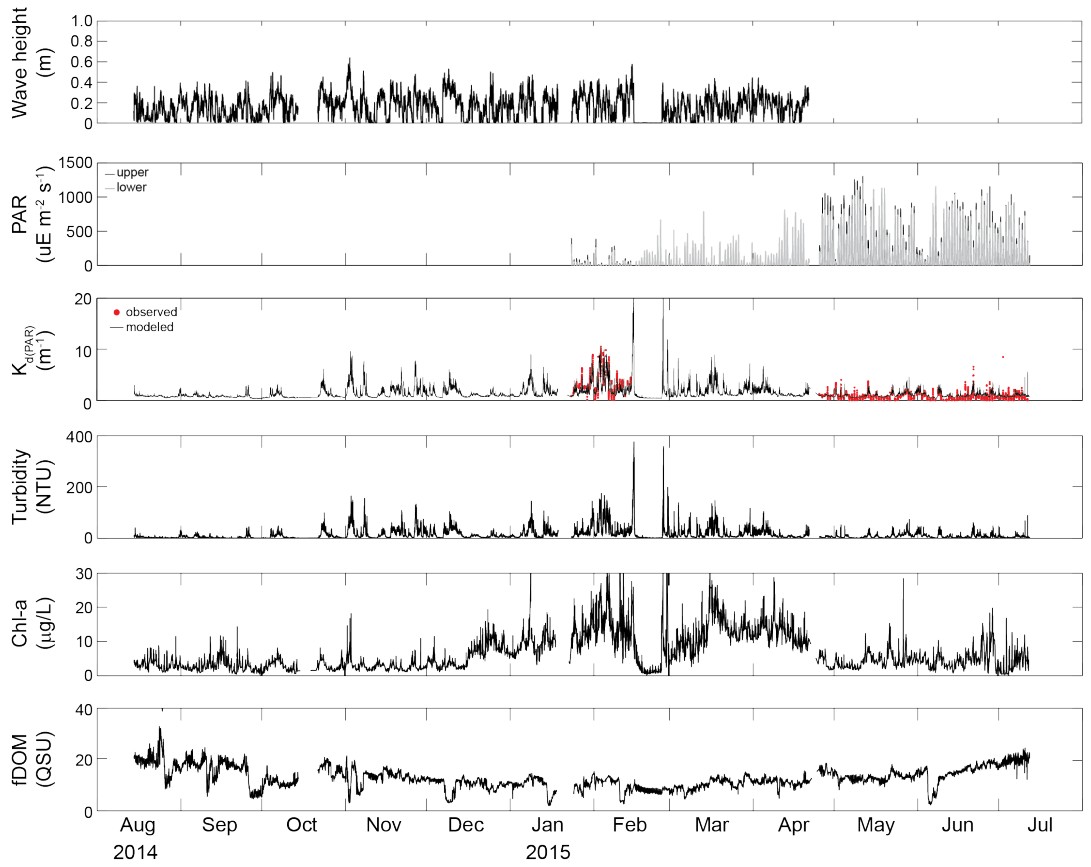

Figure 3. Time-series of wave height, upper and lower PAR, $K_{dPAR}$, turbidity, chlorophyll-a, and fDOM at vegetated shoal site CB10.


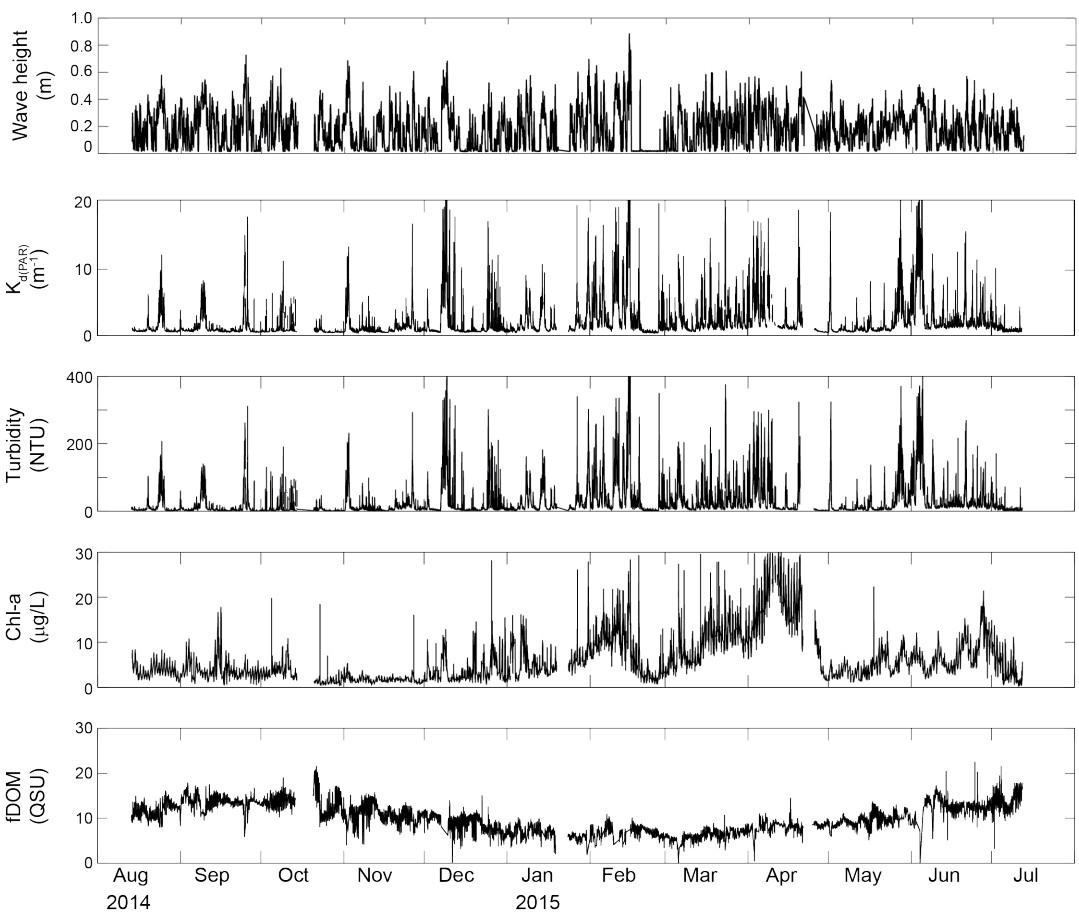

Figure 4. Time-series of wave height, modeled $K_{dPAR}$, turbidity, chlorophyll-a, and fDOM at unvegetated channel site CB06.



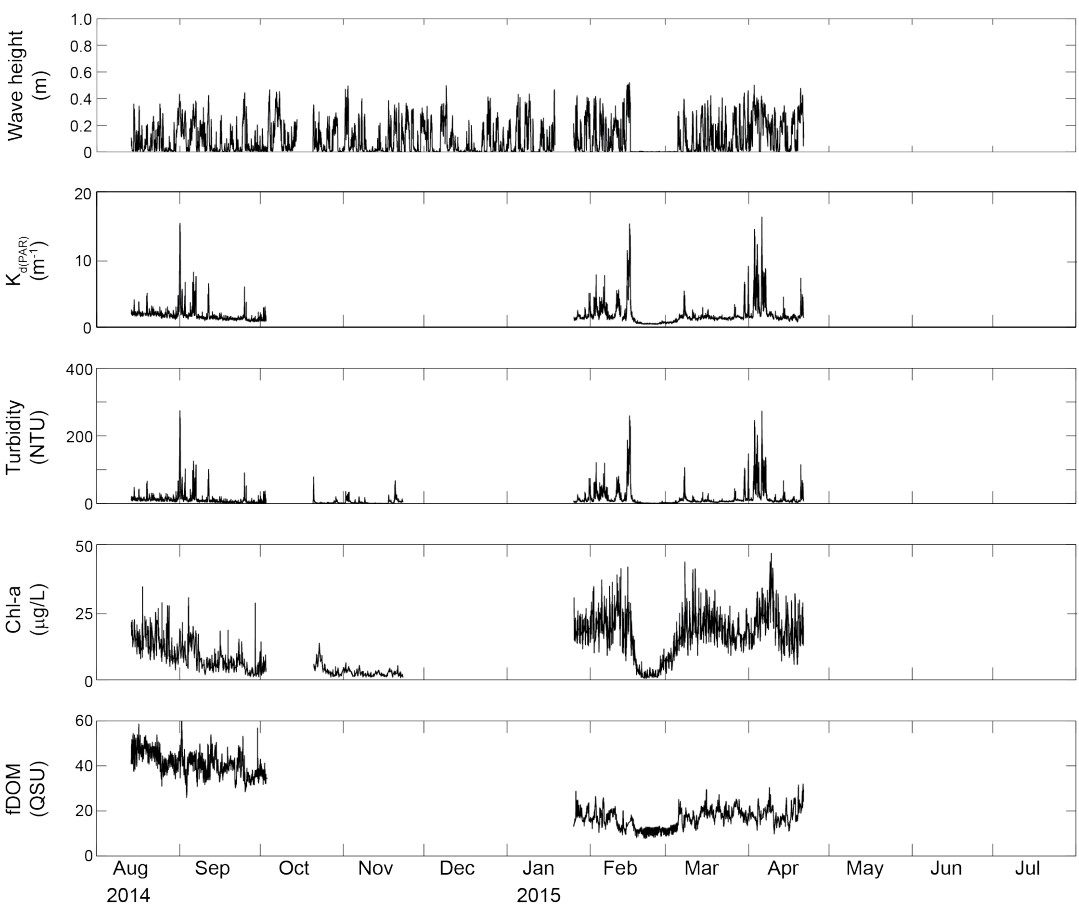

Figure 5. Time-series of wave height, modeled $K_{dPAR}$, turbidity, chlorophyll-a, and fDOM at unvegetated channel site CB11.





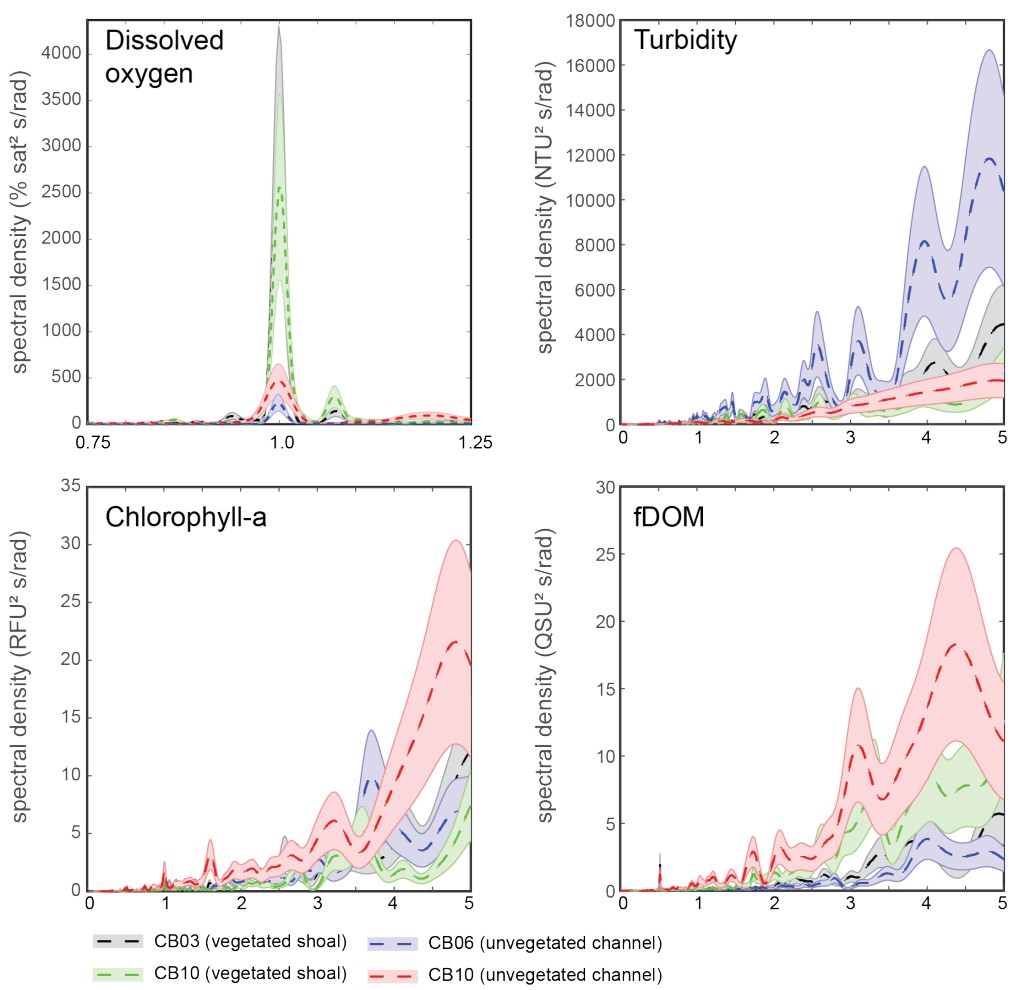

Figure 6. Spectral density estimates for dissolved oxygen, turbidity, chlorophyll-a, and fDOM at all four sites.

Shaded areas indicate 90% uncertainty bounds of spectral density estimates.



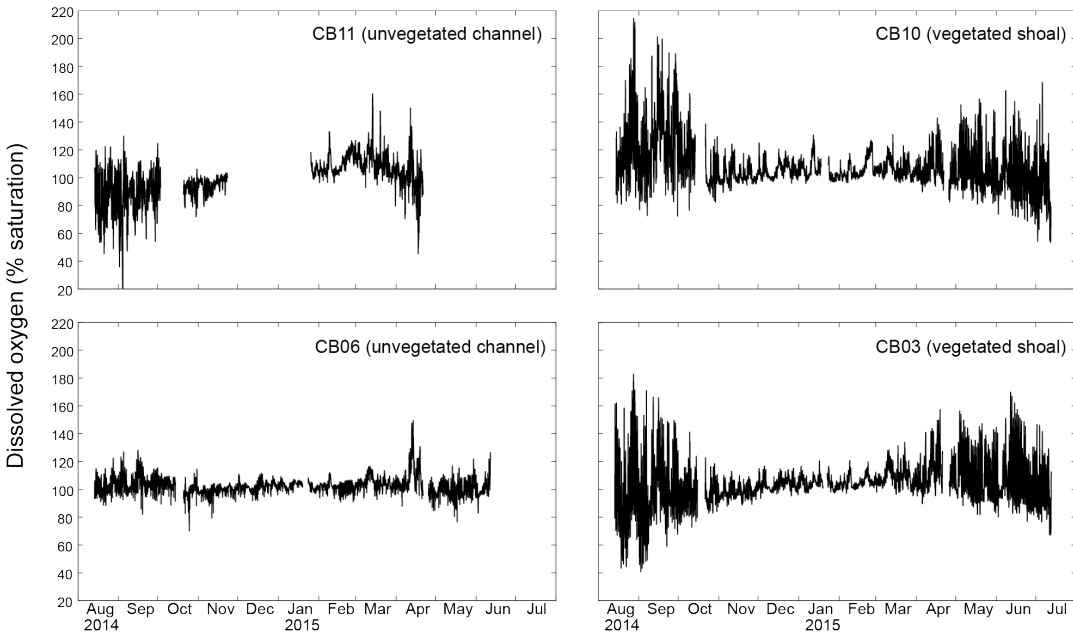

Figure 7. Time-series of dissolved oxygen from four sites.

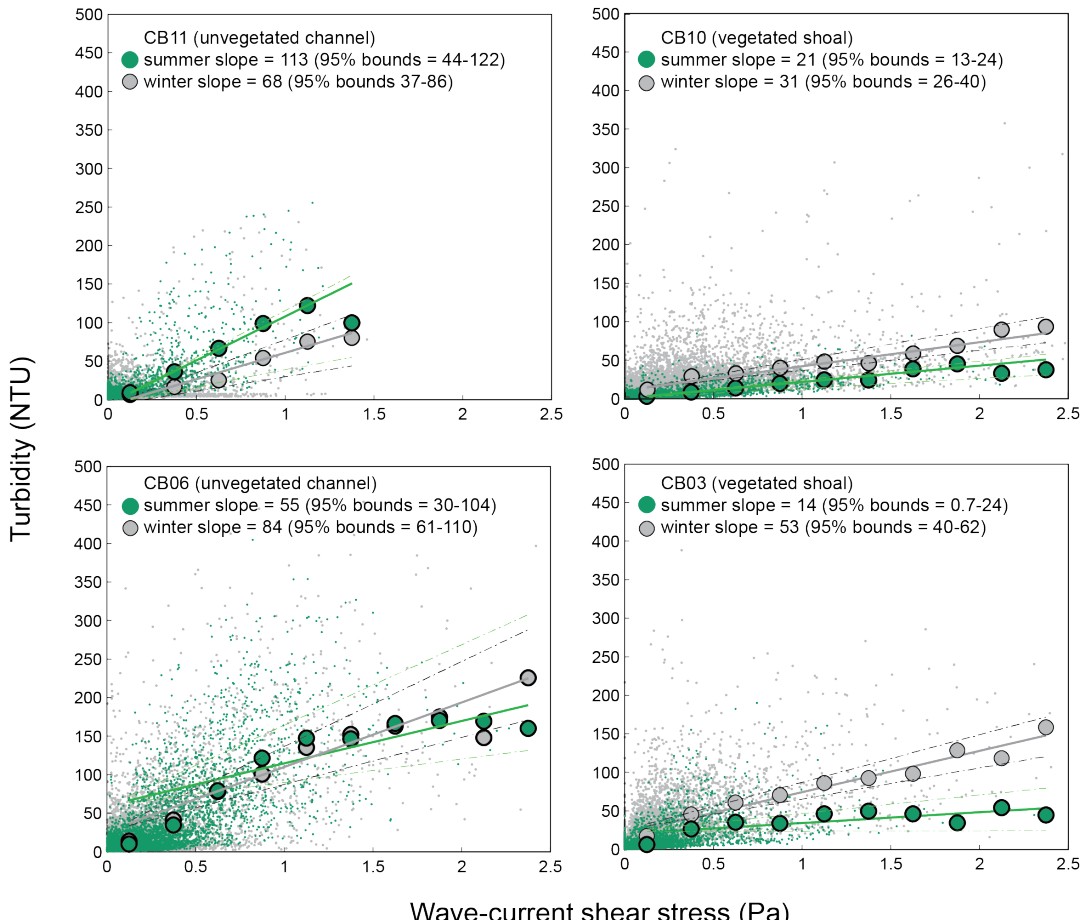

Figure 8. Relationship between combined wave-current induced bed shear stress and turbidity at four sites, with repeated-median linear regressions of bin-averaged values over summer (May-September) and winter (October-April) seasons. Larger slopes indicate a stronger relationship between shear stress and turbidity. Dashed lines

5   indicate 95% confidence bounds on slopes; bounds overlap at channel sites indicating a lack of significant difference between seasons, while bounds do not overlap at shoal sites indicating a significant seasonal difference in the stress-turbidity relationship, likely caused by vegetation.
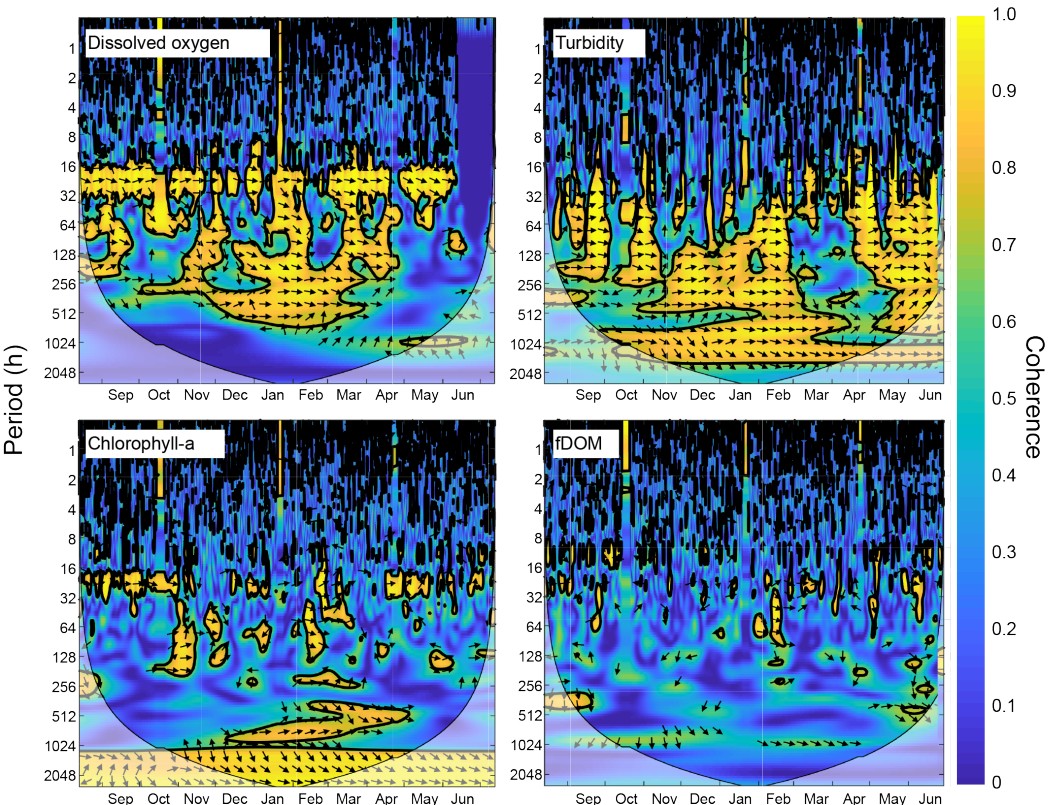

Figure 9. Wavelet coherence between time-series at sites CB03 (vegetated shoal) and CB06 (unvegetated channel), for four water quality parameters. Increased coherence at a given period indicates co-variability of the time-series at that time; for example, increased coherence at ~24 h for dissolved oxygen during most of the record indicates co-varying diel oscillations at both sites during most of the year. Direction of arrows indicates phase; arrows pointing to the right indicate that signals are in phase.

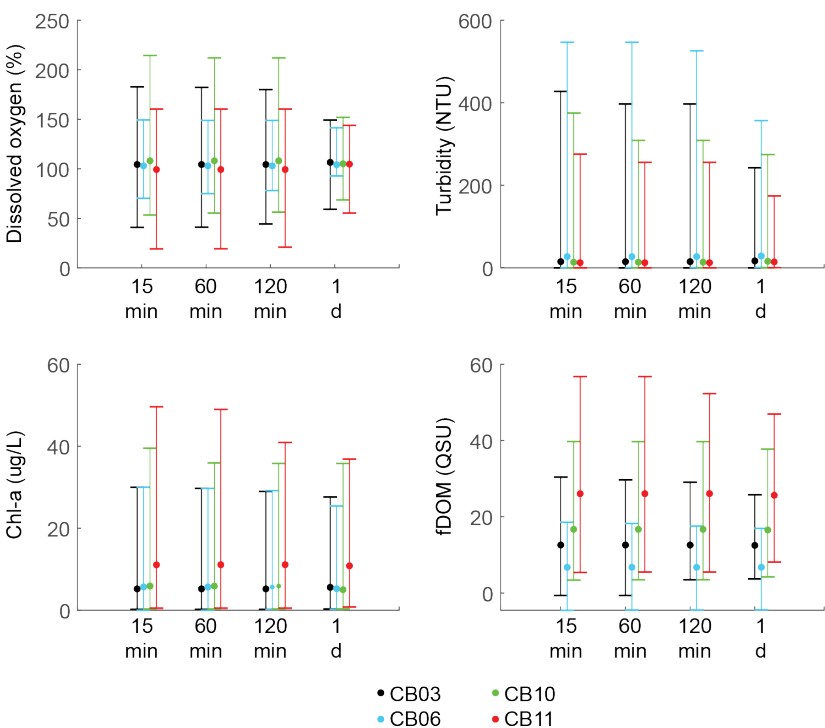

Figure 10. Mean (dots), maxima and minima (bars) for each parameter using different temporal sampling intervals (15 min, 1 h, 2 h, 24h). Spatial gradients in dissolved oxygen are most impacted by coarse temporal resolution, with spatial differences in minima and maxima largely eliminated at resolution of 1 d.





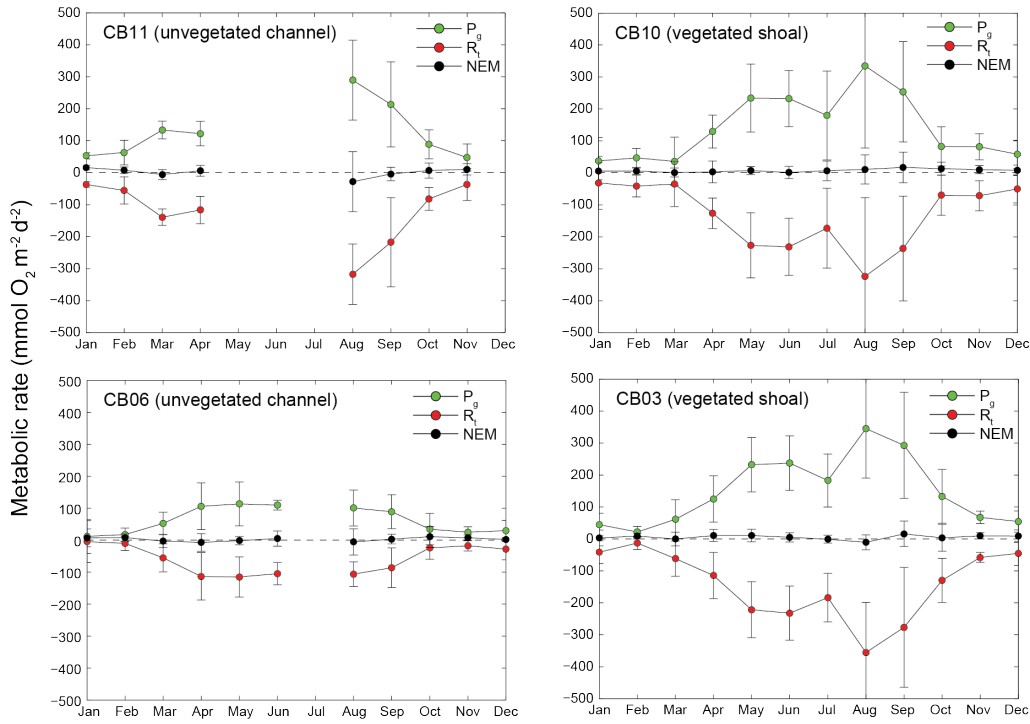

Figure 11. Monthly estimates of gross primary production ($P_g$), respiration ($R_t$), and net ecosystem metabolism (NEM) at four sites. Measurements span August 2014 through July 2015, therefore time axis begins in August 2014 and wraps back to January 2015.



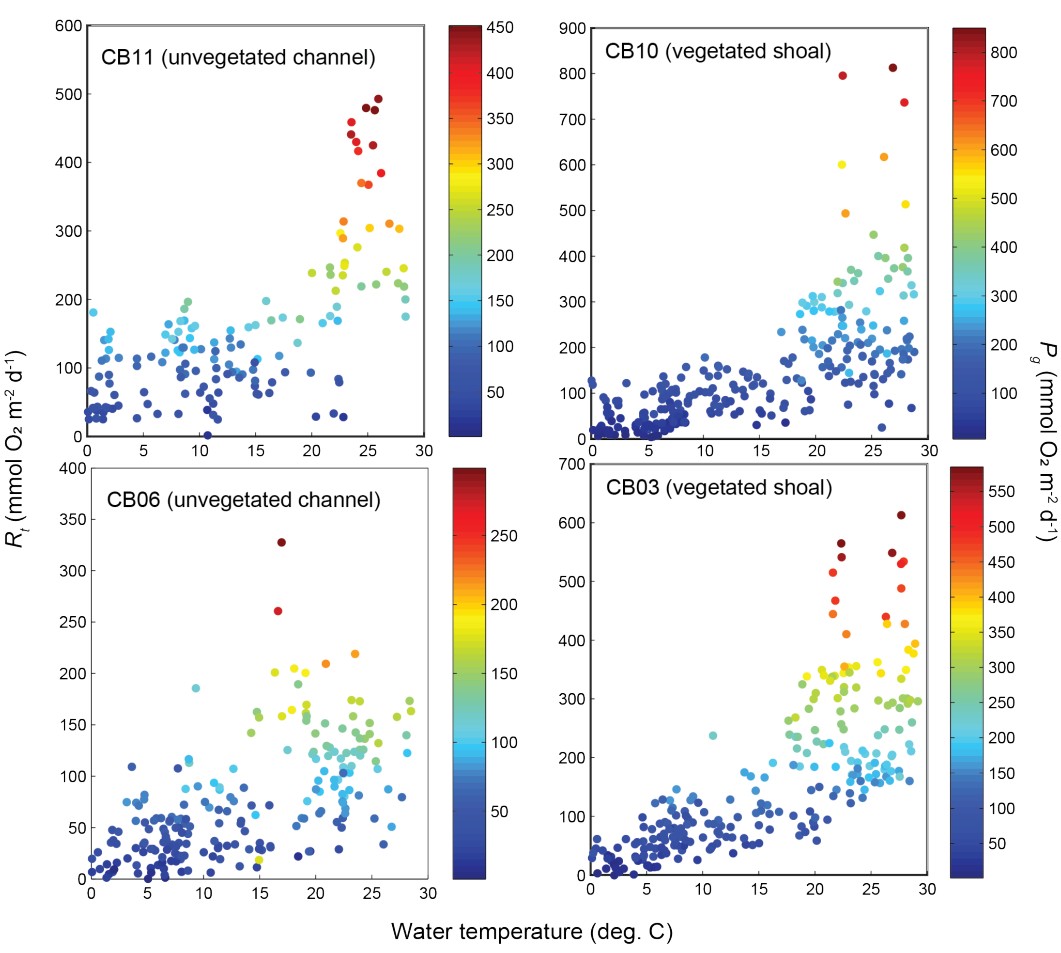

Figure 12. Relationship between respiration ($R_t$) and water temperature at four sites, with data coloration scaled to gross primary production ($P_g$).



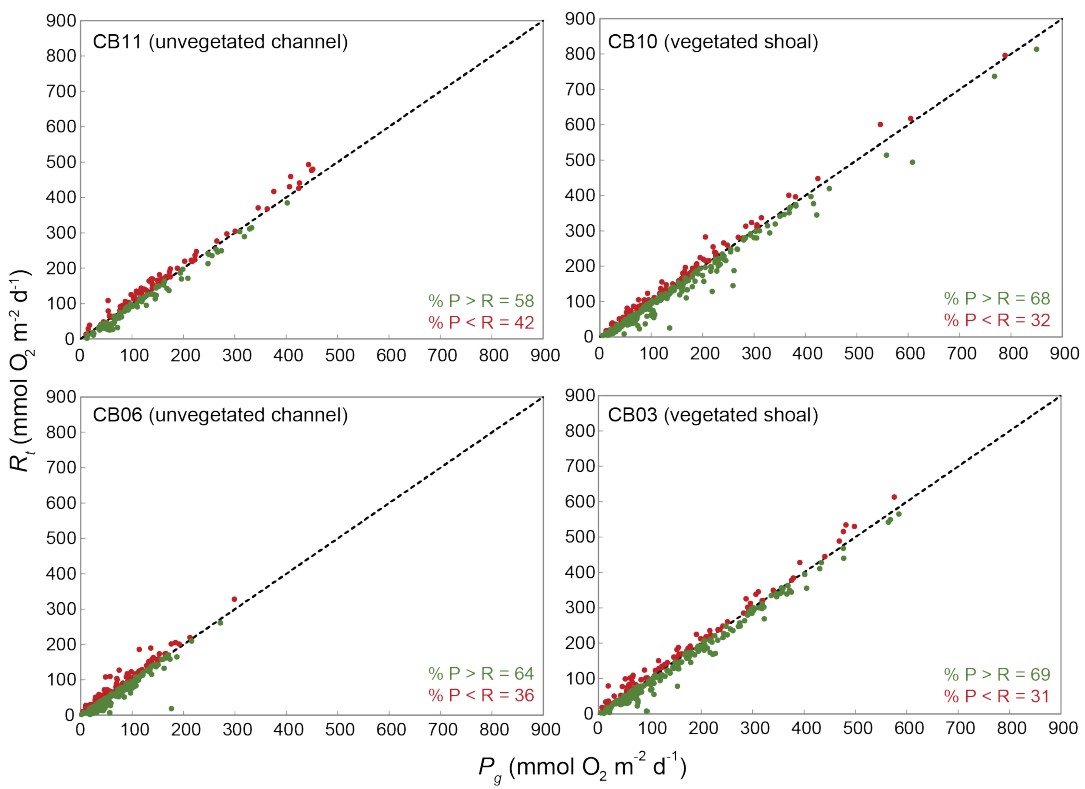

Figure 13. Relationship between gross primary production ($P_g$) and respiration ($R_t$) at four sites, line of 1:1

agreement shown.

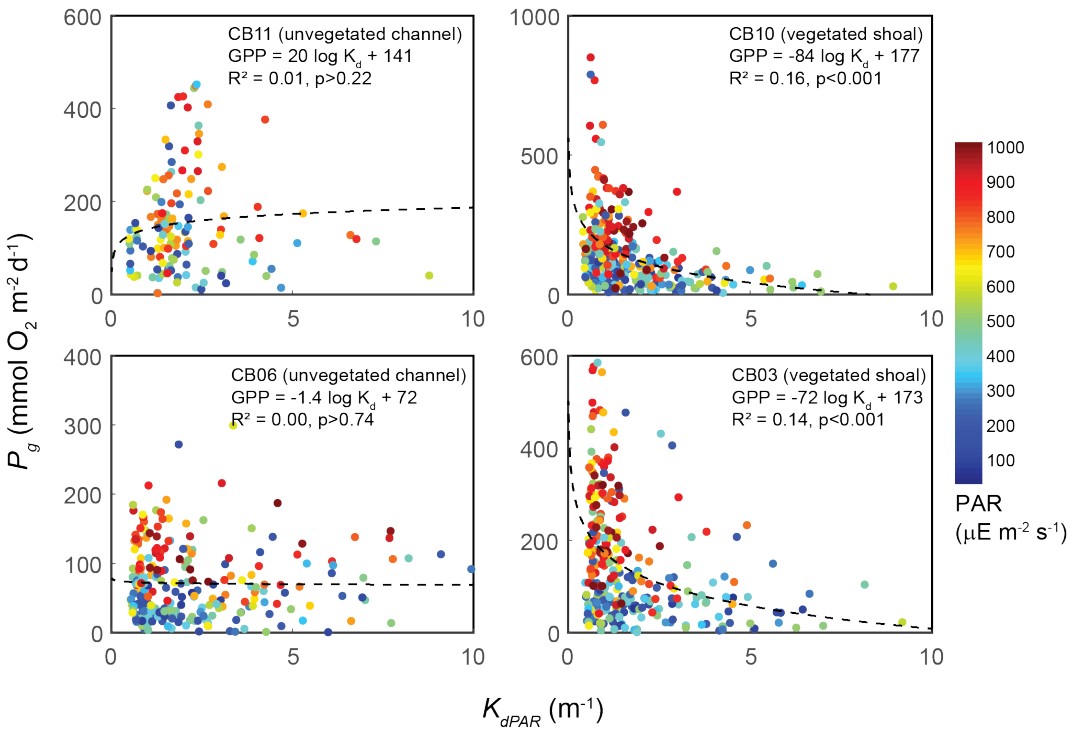

Figure 14. Relationship between light attenuation $K_{dPAR}$ and gross primary production ($P_g$), with coloration indicating surface photosynthetically active radiation (PAR, $\mu M\ m^{-2}\ s^{-1}$) measured at the weather station. Linear model was fitted to $P_g$ as a function of log ($K_{dPAR}$). The relationship between $P_g$ and $K_{dPAR}$ significant at vegetated shoal sites, but not significant at unvegetated channel sites.