# Peer review of "Spatiotemporal variability of light attenuation and net ecosystem metabolism in a back-barrier estuary"

_Ocean Science, 2019_

## Referee Comment (RC1) · Anonymous Referee #1 · 10 Feb 2020

The manuscript by Ganju et al. uses long-term high frequency measurements to quantify biogeochemical dynamics, coherence and metabolism in a shallow back-barrier estuary with focus on the role of submerged aquatic vegetation (SAV). The experimental design comprises four monitoring stations with water quality probes equipped with sensors for turbidity, chlorophyll-a, fDOM and oxygen. PAR sensors were included at two stations. It is concluded that SAV reduce sediment resuspension and thus Kd and that vegetated sites exhibited higher metabolic rates compared to un-vegetated sites. Unfortunately, these conclusions are not fully supported by the experimental design and results as explained in detail below. This said, the obtained data seem to be of high quality and may be used to provide some insights into the spatial heterogeneity

and metabolic characteristics of contrasting sites within an estuary.

Specific comments: The experimental design is not well-chosen for testing the scientific hypothesis of the paper (i..e that spatiotemporal variability/gradients in well-mixed estuaries is driven by the presence of SAV). Although there are two vegetated and two un-vegetated sites, these sites differ with respect to depth, substrate, eutrophication, etc making it impossible to attribute differences between vegetated and un-vegetated sites to SAV alone. The hypothesis/aim of the study should be revised. Also, the physical/hydro-morphological/biological characteristics incl. hydrodynamic properties of the monitoring sites as well as the rationale behind the choice of the sites should be carefully described.

It is not clear how the signal from the sensors were quality assured/rinsed for outliers, off sets, drifting etc. before use. Please describe any post processing procedure of the raw signals. The max values in table 3 don't indicate a "problem" with e.g. artificial spikes (which is somewhat surprising). However, the 0 values of turbidity, chla and fDOM seems a bit unrealistic.

Time series of chlorophyll, with high concentrations during resuspension events suggest domination of suspended microphytobenthos/dead microalgae. Describe/discus if microalgae primary production is dominated by pelagic and/or benthic microalgae.

It is concluded that the presence of SAV controls the shear stress-resuspension relation (P20, L 3). However, as the physical characteristics of especially the shallow sites suggest that microphytobenthos could be abundant, this would significantly influence the shear stress-resuspension relationship as well. Hence, SAV may not be the only/main explanation for the observed seasonal trend in shear stress-resuspension relation.

Results of significance test/SD would be highly appreciated, when presenting the results for the four sites (e.g. table 3 and text). Fx. it is states that mean (over what time period?) of chla concentration at station CB11 is twice as high as for the other stations

(P10, L11-13). It would be nice to know that the observed difference is significant.

More importantly, it is concluded that vegetated sites exhibited higher metabolic rates compared to un-vegetated sites (e.g. P20, L 6). This conclusion is not supported by the results in Table 1 where the apparent difference in Pg and Rf does not seem to be significant between site CB3, CB10 and CB11. Furthermore, Pn for all stations seems to be statistically similar.

Direct PAR measurements (and derived Kd values) are only performed at the two shallow, vegetated sites. Although it is difficult to assess the quality/performance of the model from fig. 2 and 3, it seems reasonable that the model can be used to close the gabs in the time series at these stations. However, since light attenuation is dependent on e.g. particle size it seems unjustified to apply the model for stations where it is not calibrated/validated especially since the sediment at the different stations seems to differ in particle size and quality (mud vs sandy sediments and organics enrichment).

Determining the optimal sampling frequency is a science in itself and the results regarding the influence of sampling frequency (table 3) is interesting, but as presented it seems as an unfinished story that is not properly treated in the result and discussion section. This part should be either up- or down scaled/skipped.

The present study do not examine gradients, so this term should be avoided (e.g. P1, L 13). Use "patchiness" or "spatial variability" instead.

Technical correction:

Table 3: State which time period (yearly, season, other?), min, max and mean is covering. Include SD for the mean values.

Figure 6: add x-axis titles

---

## Referee Comment (RC2) · Mario Hoppema (Referee) · 28 Feb 2020

The manuscript presents an impressive data set for four locations in an estuary, including several useful variables/parameters. The data interpretation reveals some interesting results and conclusions, which are definitely worth publishing in Ocean Science. However, in some cases there seems to be over-interpretation, for example the difference in metabolic rates between vegetated and unvegetated sites. Also the data processing should be explained in somewhat more detail.

For the net ecosystem metabolism, sometimes NEM is used, for example in Figure 11, and sometimes Pn, for example in Table 1. Please use only one single symbol or

abbreviation throughout the manuscript. Below a list with more detailed comments.

P6, L3 please define RBR D|Wave P6, L7 please define ADCP

P9, L2 "across sites across habitats" Modify piece of text? P10, L15 I would recommend not to use NPQ as abbreviation, as it is only used twice; the reader possibly has to search for it here.

P10, L19-20 "with the highest values consistently observed at CB11" I think one cannot state that, because the record at CB11 is far from complete; certainly the word "consistently" is misleading here.

P10, L20 "with lowest fDOM in the winter, possibly due to reduced biological activity" This does not sound like a good explanation. fDOM is not high during maxima of chlorophyll, which seems to indicate that it is not produced by biological activity. Moreover, the highest values seem to occur at site CB11, which receives most riverine freshwater.

P11, L14-16 "Light attenuation at site CB11, with its proximity to freshwater and nutrient sources, was highest overall and more highly influenced by chlorophyll-a and fDOM than at other sites." This is hard to believe when inspecting Figures 2-5. Moreover, there is much less data available for this site than for the others.

P12. L13-14 "between May and October" My view of the figures says that this should be "between May and September".

P12, L14 "during November to April" I was say "during November to March" P12, L19-20 "but instances of net autotrophy (Pg > Rt) occurred nearly 70% of the time at the vegetated sites" When I view Figure 11, neither autotrophy nor heterotrophy are statistically significant. This should be mentioned here.

P13, L10 at 1-7 day P13, L23-24 "The peak in spectral density was 30% higher at CB03 than CB10" This is really hard to see, if at all, in Fig. 6. Possibly the authors could add a comment to Figure 6 (Diss Oxygen) that the curve of CB3 lies under the

one of CB10.

P17, L26 where the canonical C:N (I suggest to add canonical, since this is the well-known Redfield ratio based on data from many locations)

P19, L1-2 "In fact, modest net autotrophy prevailed during the summer season at vegetated sites but not at un-vegetated sites." This does not follow from the data in Figure 11, where NEM is around zero all through the year. See also comment above. Actually I am surprised by the uncertainty interval of NEM, which should have been formed from subtracting two larger terms. Because of this, I would expect it to be clearly larger and not smaller than the uncertainty of both of the terms.

References P22, L4 The usual abbreviation is: Limnol. Oceanogr. P22, L7 What is this? Ref. No. [UMCES]CBL 04-105a? Report? P22, L23 The usual abbreviation is: Estuar. Coast. Shelf Sci., P23, L16 change: Fremantle P23, L22 The usual abbreviation is: Estuar. Coast. Shelf Sci., P24, L5 The usual abbreviation is: Estuar. Coast. Shelf Sci., P24, L16 If the journal consists of one word, the full name should be given: Biogeosciences P25, L8 Modify journal name: Mar. Ecol. Prog. Ser. P25, L16 Add pages and doi: S3-S16. doi:10.1890/05-0800.1 P25, L23 The usual abbreviation is: Limnol. Oceanogr. P26, L7 The usual abbreviation is: Limnol. Oceanogr. P26, L16 Add volume and pages: Mar. Geol., 404, 1-14 P26, L18 The usual abbreviation is: Limnol. Oceanogr. P26, L22 Use full journal name: Biogeosciences P27, L2 The usual abbreviation is: Estuar. Coast. Shelf Sci., P27, L12 Please give more info on this publication; report, website? P27, L14 dito P27, L16 The usual abbreviation is: Limnol. Oceanogr. P27, L18 Ocean Mod. P27, L20 Please add page numbers

Table 1 Please spell out standard deviation

Figure 1 Please also define ADCP and CBWS . Figure 6 I think CB10 in red under the figure panels must be CB11 (unvegetated) Please indicate panels (a) − (d), and then also in the caption. Please explain the x-axis; in the main text the authors talk about "peak at 12.4 h", but this is not reflected in the figure.

Figure 7 It is not clear to me how the oxygen saturation could be 130% in the middle of winter (e.g. at CB11).

Figure 9 Please indicate panels (a) – (d), and then also in the caption. Please explain what the bottom structures in the panels mean, and why the data are shown despite these structures.

---

## Author Comment (AC1) · 24 Mar 2020

Response to reviewers, comments in normal text, **responses in bold**

Anonymous Referee #1

The manuscript by Ganju et al. uses long-term high frequency measurements to quantify biogeochemical dynamics, coherence and metabolism in a shallow back-barrier estuary with focus on the role of submerged aquatic vegetation (SAV). The experimental design comprises four monitoring stations with water quality probes equipped with sensors for turbidity, chlorophyll-a, fDOM and oxygen. PAR sensors were included at two stations. It is concluded that SAV reduce sediment resuspension and thus Kd and that vegetated sites exhibited higher metabolic rates compared to un-vegetated sites.

Unfortunately, these conclusions are not fully supported by the experimental design and results as explained in detail below. This said, the obtained data seem to be of high quality and may be used to provide some insights into the spatial heterogeneity and metabolic characteristics of contrasting sites within an estuary.

**Thank you for your positive comments. We will rephrase throughout to indicate that the turbidity-shear stress relationships indicate a seasonal pattern that may be related to bed stabilization by microphytobenthos and increased SAV aboveground biomass in the summer.**

Specific comments: The experimental design is not well-chosen for testing the scientific hypothesis of the paper (i..e that spatiotemporal variability/gradients in well-mixed estuaries is driven by the presence of SAV). Although there are two vegetated and two un-vegetated sites, these sites differ with respect to depth, substrate, eutrophication, etc making it impossible to attribute differences between vegetated and un-vegetated sites to SAV alone. The hypothesis/aim of the study should be revised.

**We will revise the hypothesis as follows:**

> **"We hypothesize that even in shallow, well-mixed estuaries there is strong spatiotemporal variability in ecosystem metabolism due to benthic and water column properties, and ensuing feedbacks to sediment resuspension, light attenuation, and primary production."**

Also, the physical/hydro-morphological/biological characteristics incl. hydrodynamic properties of the monitoring sites as well as the rationale behind the choice of the sites should be carefully described.

**We will add details on the hydrodynamic climate (tide range, subtidal water level fluctuations, velocity), lithology, and organic matter content of the substrate. We will also add rationale behind site selection.**

It is not clear how the signal from the sensors were quality assured/rinsed for outliers, off sets, drifting etc. before use. Please describe any post processing procedure of the raw signals. The max values in table 3 don't indicate a "problem" with e.g. artificial spikes (which is somewhat surprising). However, the 0 values of turbidity, chla and fDOM seems a bit unrealistic.

**The USGS-required data report, cited in the manuscript, contains these details, and we will refer to it in more detail. For brevity we will add the essential details to the text.**

Time series of chlorophyll, with high concentrations during resuspension events suggest domination of suspended microphytobenthos/dead microalgae.

**Agreed, we failed to note that where we discussed peaks in chlorophyll during winter resuspension events in the Results section. We will add this detail.**

Describe/discus if microalgae primary production is dominated by pelagic and/or benthic microalgae.

**We do not have direct estimates of microalgal production partitioned between sediments and the water column at any of our sites. At the two SAV-dominated sites, the sediment surface is nearly covered by SAV and macroalgae, so we expect that benthic microalgal production is low. At CB11, the water is very turbid and kd values indicate that very little light reaches the bottom, so we can presume that water-column microalgae are dominant. For the other channel site, we cannot easily constrain the partitioning between the two environments. We will add these details to the manuscript.**

It is concluded that the presence of SAV controls the shear stress-resuspension relation (P20, L 3). However, as the physical characteristics of especially the shallow sites suggest that microphytobenthos could be abundant, this would significantly influence the shear stress-resuspension relationship as well. Hence, SAV may not be the only/main explanation for the observed seasonal trend in shear stress-resuspension relation.

**Agreed, the most general explanation is that depth controls the availability of light for both SAV and benthic algae; and the presence of both to some degree stabilize the bed in summer more than winter. At the deep sites, no such relationship exists, indicating no change in benthic characteristics. However, it is likely that in a vegetated sandy substrate, the seagrass would dominate bed stabilization over benthic microalgae given shading and coarser sediment. In fact, the data of Ellis et al. indicate organic matter percentages of less than 2% on the sandy shoals. However, we will indicate that benthic algae may contribute to bed stabilization in the summer, and likely account for chlorophyll resuspension signals in winter.**

**Ellis, A.M., Marot, M.E., Wheaton, C.J., Bernier, J.C., and Smith, C.G., 2015, A seasonal comparison of surface sediment characteristics in Chincoteague Bay, Maryland and Virginia, USA: U.S. Geological Survey Open-File Report 2015-1219, http://dx.doi.org/10.3133/ofr20151219.**

Results of significance test/SD would be highly appreciated, when presenting the results for the four sites (e.g. table 3 and text). Fx. it is states that mean (over what time period?) of chla concentration at station CB11 is twice as high as for the other stations (P10, L11-13). It would be nice to know that the observed difference is significant.

**In the revision, we will test significance for all parameters by comparing overlap in means and standard deviations.**

More importantly, it is concluded that vegetated sites exhibited higher metabolic rates compared to un-vegetated sites (e.g. P20, L 6). This conclusion is not supported by the results in Table 1 where the apparent difference in Pg and Rf does not seem to be significant between site CB3, CB10 and CB11. Furthermore, Pn for all stations seems to be statistically similar.

**We believe that the data clearly show that Pg and Rf are higher at sites CB3, CB10 and CB11 – which is what we refer to with the term "metabolic rates". We also state in the text that CB11 is similar to CB3 and CB10 because CB11 is a highly eutrophic site with high chlorophyll-a and water-column phytoplankton production. We agree than Pn is statistically similar, given that Pg and Rf tend to**

**balance each other despite higher gross rates of Pg and Rf.**

Direct PAR measurements (and derived Kd values) are only performed at the two shallow, vegetated sites. Although it is difficult to assess the quality/performance of the model from fig. 2 and 3, it seems reasonable that the model can be used to close the gabs in the time series at these stations. However, since light attenuation is dependent on e.g. particle size it seems unjustified to apply the model for stations where it is not calibrated/validated especially since the sediment at the different stations seems to differ in particle size and quality (mud vs sandy sediments and organics enrichment).

**The light model assessment for sites CB03 and CB10 is shown in supplementary Figure S2. Nonetheless, to test the sensitivity, we ran the light model with variation in the particle backscatter ratio ($b_{bx}$) which is particle size dependent. Please note that an increase in flocculated sediments would increase this ratio, thereby increasing the light attenuation and causing a larger difference in light attenuation between the sandy, vegetated sites and the muddy, unvegetated sites. Modifying the value from 0.017 to 0.025 (47% increase; 0.025 value was used for a different mud-dominated estuary by Ganju et al., 2014) increased the median light attenuation at CB06 by 17% (from 1.35 to 1.58 m$^{-1}$) and at CB11 by 11% (from 1.67 to 1.86). We will add this analysis to the revision.**

Determining the optimal sampling frequency is a science in itself and the results regarding the influence of sampling frequency (table 3) is interesting, but as presented it seems as an unfinished story that is not properly treated in the result and discussion section. This part should be either up- or down scaled/skipped.

**We believe that the concept that coarse temporal sampling masks spatial variability is an important one given the common daily sampling approach for dissolved oxygen in many impaired estuaries. We will highlight this concept with examples from the literature in our revision.**

The present study do not examine gradients, so this term should be avoided (e.g. P1, L 13). Use "patchiness" or "spatial variability" instead.

**Agreed, we will revise to "variability".**

Technical correction: Table 3: State which time period (yearly, season, other?), min, max and mean is covering. Include SD for the mean values. Figure 6: add x-axis titles

**We will correct these.**

Mario Hoppema (Referee) mario.hoppema@awi.de

The manuscript presents an impressive data set for four locations in an estuary, including several useful variables/parameters. The data interpretation reveals some interesting results and conclusions, which are definitely worth publishing in Ocean Science.

However, in some cases there seems to be over-interpretation, for example the difference in metabolic rates between vegetated and unvegetated sites.

**In addition to our response to Reviewer #1 above, we would like to point out that the rates are statistically significant between CB3 and CB10 compare with CB6 – which is the main distinction we discussed between vegetated and unvegetated. We will add statistical significance in the revision.**

Also the data processing should be explained in somewhat more detail.

**We will revise following this comment and Reviewer 1's comment (see above).**

For the net ecosystem metabolism, sometimes NEM is used, for example in Figure 11, and sometimes Pn, for example in Table 1. Please use only one single symbol or abbreviation throughout the manuscript.

**We will correct this.**

Below a list with more detailed comments.

P6, L3 please define RBR D|Wave

**We will revise this to "RBR Virtuoso D|Wave pressure recorder".**

P6, L7 please define ADCP

**We will revise this to "acoustic Doppler current profiler (ADCP)".**

P9, L2 "across sites across habitats" Modify piece of text?

**We will revise to across sites and across habitats.**

P10, L15 I would recommend not to use NPQ as abbreviation, as it is only used twice; the reader possibly has to search for it here.

**We will revise to "non-photochemical quenching".**

P10, L19-20 "with the highest values consistently observed at CB11" I think one cannot state that, because the record at CB11 is far from complete; certainly the word "consistently" is misleading here.

**We will revise to "during periods when data were available at CB11".**

P10, L20 "with lowest fDOM in the winter, possibly due to reduced biological activity" This does not sound like a good explanation. fDOM is not high during maxima of chlorophyll, which seems to indicate that it is not produced by biological activity. Moreover, the highest values seem to occur at site CB11, which receives most riverine freshwater.

**This was meant to suggest that reduced activity in the surrounding watershed and water bodies where DOM is produced, and then exported to the estuary with freshwater. We will clarify in the revision.**

P11,L14-16"Light attenuation at site CB11,with its proximity to freshwater and nutrient sources, was highest overall and more highly influenced by chlorophyll-a and fDOM than at other sites." This is hard to believe when inspecting Figures 2-5. Moreover, there is much less data available for this site than for the others.

**We will revise to indicate that this is only true during periods of overlap.**

P12. L13-14 "between May and October" My view of the figures says that this should be "between May and September".

**We will revise.**

P12, L14 "during November to April" I was say "during November to March"

**We will revise.**

P12, L1920 "but instances of net autotrophy (Pg > Rt) occurred nearly 70% of the time at the vegetated sites" When I view Figure 11, neither autotrophy nor heterotrophy are statistically significant. This should be mentioned here.

**It is a fair comment to suggest that we did not include statistical analyses to "prove" the existence of net autotrophy, but rather expressed the frequency of net autotrophic conditions. We will mention this consideration of the analysis in the revised discussion.**

P13, L10 at 1-7 day

**We will revise.**

P13, L23-24 "The peak in spectral density was 30% higher at CB03 than CB10" This is really hard to see, if at all, in Fig. 6. Possibly the authors could add a comment to Figure 6 (Diss Oxygen) that the curve of CB3 lies under the one of CB10.

**We will include a comment in the caption or modify the figure for clarity.**

P17, L26 where the canonical C:N (I suggest to add canonical, since this is the well known Redfield ratio based on data from many locations)

**We will revise.**

P19, L1-2 "In fact, modest net autotrophy prevailed during the summer season at vegetated sites but not at un-vegetated sites." This does not follow from the data in Figure 11, where NEM is around zero all through the year. See also comment above. Actually I am surprised by the uncertainty interval of NEM, which should have been formed frrom subtracting two larger terms. Because of this, I would expect it to be clearly larger and not smaller than the uncertainty of both of the terms.

**If you compare NEM to Pg and Rt on an absolute basis, we agree that NEM is near zero. But our statement simply refers to the mean monthly value of NEM, which although much smaller than Pg and Rt, is still greater than zero at the vegetated sites during some seasons, which we define as autotrophic. The error bars around NEM represent the standard deviation of the monthly mean estimates of NEM, which the reviewer is correct in that they are derived as the difference between Pg and Rt. But these difference calculations are performed on a daily basis, so the error bars represent the total variation of all ~30 daily Pg, Rt, and NEM rates and thus do not represent uncertainty carried over from each individual NEM calculation.**

References

P22, L4 The usual abbreviation is: Limnol. Oceanogr.

P22, L7 What is this? Ref. No. [UMCES]CBL 04-105a? Report?

P22, L23 The usual abbreviation is: Estuar. Coast. Shelf Sci.,

P23, L16 change: Fremantle

P23, L22 The usual abbreviation is: Estuar. Coast. Shelf Sci.,

P24, L5 The usual abbreviation is: Estuar. Coast. Shelf Sci.,

P24, L16 If the journal consists of one word, the full name should be given: Biogeosciences

P25, L8 Modify journal name: Mar. Ecol. Prog. Ser.

P25, L16 Add pages and doi: S3-S16. doi:10.1890/05-0800.1

P25, L23 The usual abbreviation is: Limnol. Oceanogr.

P26, L7 The usual abbreviation is: Limnol. Oceanogr.

P26, L16 Add volume and pages: Mar. Geol., 404, 1-14

P26, L18 The usual abbreviation is: Limnol. Oceanogr.

P26, L22 Use full journal name: Biogeosciences

P27, L2 The usual abbreviation is: Estuar. Coast. Shelf Sci.,

P27, L12 Please give more info on this publication; report, website?

P27, L14 dito

P27, L16 The usual abbreviation is: Limnol. Oceanogr.

P27, L18 Ocean Mod.

P27, L20 Please add page numbers

**We will correct all reference errors.**

Table 1 Please spell out standard deviation

**We will revise.**

Figure 1 Please also define ADCP and CBWS .

**We will revise.**

Figure 6 I think CB10 in red under the figure panels must be CB11 (unvegetated) Please indicate panels (a) – (d), and then also in the caption. Please explain the x-axis; in the main text the authors talk about "peak at 12.4 h", but this is not reflected in the figure.

**We will revise.**

Figure 7 It is not clear to me how the oxygen saturation could be 130% in the middle of winter (e.g. at CB11).

**The periods where oxygen saturation exceed 130% do not truly occur in winter, which we would define as mid-December to mid-March. The exception to this is during a brief period in February and early March at CB 11, where chlorophyll-a exceeded 25 ug/L indicating that phytoplankton biomass was high and presumably, primary production rates.**

Figure 9 Please indicate panels (a) – (d), and then also in the caption. Please explain what the bottom structures in the panels mean, and why the data are shown despite these structures.

**We will revise; the shaded areas are zones of the parameter space that are not statistically robust. We will eliminate these in the revision.**

---

## Author Response (AR1)

Response to reviewers, comments in normal text, **responses in bold. Line numbers refer to the final version of the manuscript.**

Anonymous Referee #1

The manuscript by Ganju et al. uses long-term high frequency measurements to quantify
5   biogeochemical dynamics, coherence and metabolism in a shallow back-barrier estuary with focus on the role of submerged aquatic vegetation (SAV). The experimental design comprises four monitoring stations with water quality probes equipped with sensors for turbidity, chlorophyll-a, fDOM and oxygen. PAR sensors were included at two stations. It is concluded that SAV reduce sediment resuspension and thus Kd and that vegetated sites exhibited higher metabolic rates compared to un-vegetated sites.

10   Unfortunately, these conclusions are not fully supported by the experimental design and results as explained in detail below. This said, the obtained data seem to be of high quality and may be used to provide some insights into the spatial heterogeneity and metabolic characteristics of contrasting sites within an estuary.

**Thank you for your positive comments. We have rephrased throughout to indicate that the turbidity-**
15   **shear stress relationships indicate a seasonal pattern that may be related to bed stabilization by microphytobenthos and increased SAV aboveground biomass in the summer.**

Specific comments: The experimental design is not well-chosen for testing the scientific hypothesis of the paper (i..e that spatiotemporal variability/gradients in well-mixed estuaries is driven by the presence of SAV). Although there are two vegetated and two un-vegetated sites, these sites differ with respect to
20   depth, substrate, eutrophication, etc making it impossible to attribute differences between vegetated and un-vegetated sites to SAV alone. The hypothesis/aim of the study should be revised.

**We have revised the hypothesis on P1, 11-14 as follows:**

> **"We hypothesize that even in shallow, well-mixed estuaries there is strong spatiotemporal variability in ecosystem metabolism due to benthic and water column properties, and ensuing**
25       **feedbacks to sediment resuspension, light attenuation, and primary production."**

Also, the physical/hydro-morphological/biological characteristics incl. hydrodynamic properties of the monitoring sites as well as the rationale behind the choice of the sites should be carefully described.

**We have added details on the hydrodynamic climate (tide range and subtidal water level fluctuations), lithology, and organic matter content of the substrate; we have also added the rationale**
30   **behind site selection, these changes are on P5, 16-27.**

It is not clear how the signal from the sensors were quality assured/rinsed for outliers, off sets, drifting etc. before use. Please describe any post processing procedure of the raw signals. The max values in table 3 don't indicate a "problem" with e.g. artificial spikes (which is somewhat surprising). However, the 0 values of turbidity, chla and fDOM seems a bit unrealistic.

35   **The USGS-required data report, cited in the manuscript, contains these details, and we have added relevant details from it on P7, 9-20:**

> **"Quality control and quality assurance checks were performed on all data and are described in detail in the data report (Suttles et al., 2017). These included removal of obvious spikes in**

**individual parameter time series using either a recursive filter or median filter technique, in which values that changed from one time point to the next by more than a set threshold were flagged and replaced with a fill value (e.g. -9999). Pre-deployment calibrations and post-deployment check were performed on all EXO2 sensors following YSI procedures outlined in**

5     **the EXO User Manual (Revision F). Linear corrections were obtained from either post-deployment calibration checks or differences between in situ values from a fouled sensor and from a cleaned, calibrated sensor at the beginning of the next deployment. If data were obviously fouled and corrections were not possible, the data were replaced with fill values. WET Labs ECO-PARSB instruments were checked to verify counts were above their "dark"**

10    **count (low-count) thresholds; therefore legitimate data collected during night were also discarded. All subsurface pressure data were corrected for changes in atmospheric pressure by using local barometric pressure data, from the meteorological station at CBWS when available, to give a more accurate representation of pressure caused by the overlying water."**

**Zero values in certain parameters are common given the multi-point sensor calibrations over large**

15    **ranges of variability. It is nearly impossible to accurately capture extremely high values and have sensitivity at the low range. There were ice-cover events during our campaign that nearly shut down advection, resuspension, and production. Values of many parameters were  zero during this time.**

Time series of chlorophyll, with high concentrations during resuspension events suggest domination of suspended microphytobenthos/dead microalgae.

20    **Agreed, we failed to note that where we discussed peaks in chlorophyll during winter resuspension events in the Results section. We have added this on P11, 3-5.**

Describe/discus if microalgae primary production is dominated by pelagic and/or benthic microalgae.

**We do not have direct estimates of microalgal production partitioned between sediments and the water column at any of our sites, and have added the following on P14, 17-21 to constrain this**

25    **uncertainty:**

**"Though we do not have direct estimates of microalgal production partitioned between sediments and the water column at any of our sites, the sediment surface is nearly covered by SAV and macroalgae at the vegetated sites, therefore we expect that benthic microalgal production is low. At site CB11, $K_{dPAR}$ values indicate that very little light reaches the bottom,**

30    **so we presume that water-column microalgae are dominant. For site CB06, we cannot constrain the partitioning between the two environments."**

It is concluded that the presence of SAV controls the shear stress-resuspension relation (P20, L 3). However, as the physical characteristics of especially the shallow sites suggest that microphytobenthos could be abundant, this would significantly influence the shear stress-resuspension relationship as well.

35    Hence, SAV may not be the only/main explanation for the observed seasonal trend in shear stress-resuspension relation.

**Agreed, the most general explanation is that depth controls the availability of light for both SAV and benthic algae; and the presence of both to some degree stabilize the bed in summer more than winter. At the deep sites, no such relationship exists, indicating no change in benthic characteristics.**

40    **However, it is likely that in a vegetated sandy substrate, the seagrass would dominate bed stabilization over benthic microalgae given shading and coarser sediment. In fact, the data of Ellis et**

al. (2015; cited now in the manuscript) indicate organic matter percentages of less than 2% on the sandy shoals. However, we have indicated that benthic algae may contribute to bed stabilization in the summer, and likely accounts for chlorophyll resuspension signals in winter.

**Ellis, A.M., Marot, M.E., Wheaton, C.J., Bernier, J.C., and Smith, C.G., 2015, A seasonal comparison of surface sediment characteristics in Chincoteague Bay, Maryland and Virginia, USA: U.S. Geological Survey Open-File Report 2015-1219, http://dx.doi.org/10.3133/ofr20151219.**

Results of significance test/SD would be highly appreciated, when presenting the results for the four sites (e.g. table 3 and text). Fx. it is states that mean (over what time period?) of chla concentration at station CB11 is twice as high as for the other stations (P10, L11-13). It would be nice to know that the observed difference is significant.

**We have tested significance for all parameters by comparing overlap in means and standard deviations. Based on overlap of the means and standard deviations, the only statistically significant difference between means is for mean production ($P_g$) at sites CB03 and CB06, and fDOM at site CB10 and CB11 (relative to site CB06). We have added these details in the text and table captions. However in a tidal system, differences between means are rarely significant compared to the tidally varying values which are substantially different. We have reworded where necessary to stress whether we are discussing peak, mean, or tidal-timescale values.**

More importantly, it is concluded that vegetated sites exhibited higher metabolic rates compared to un-vegetated sites (e.g. P20, L 6). This conclusion is not supported by the results in Table 1 where the apparent difference in Pg and Rf does not seem to be significant between site CB3, CB10 and CB11. Furthermore, Pn for all stations seems to be statistically similar.

**Again, in line with the comment above, the diurnal variability in $P_g$ and $R_f$ are higher at sites CB3, CB10 and CB11 – which is what we refer to with the term "metabolic rates"—though the means are not statistically different (except for between site CB03 and site CB06). We also state in the text that CB11 is similar to CB3 and CB10 because CB11 is a highly eutrophic site with high chlorophyll-a and water-column phytoplankton production. In terms of mean values, the only statistically significant difference is between sites CB03 and CB06, in terms of $P_g$. We have noted that in the caption and text.**

Direct PAR measurements (and derived Kd values) are only performed at the two shallow, vegetated sites. Although it is difficult to assess the quality/performance of the model from fig. 2 and 3, it seems reasonable that the model can be used to close the gabs in the time series at these stations. However, since light attenuation is dependent on e.g. particle size it seems unjustified to apply the model for stations where it is not calibrated/validated especially since the sediment at the different stations seems to differ in particle size and quality (mud vs sandy sediments and organics enrichment).

**The light model assessment for sites CB03 and CB10 is shown in supplementary Figure S2. Nonetheless, to test the sensitivity, we ran the light model with variation in the particle backscatter ratio ($b_{bx}$) which is particle size dependent, and added the results on P12, 14-22:**

> **"Sediments at site CB06 and CB11 tend to be finer (Ellis et al., 2015) and may be more susceptible to flocculation, which would induce error in the light model. Therefore we ran the light model with variation in the maximum particle backscatter ratio ($\underline{b_{bx}}$) which is particle size dependent. An increase in flocculated suspended-sediment would increase this ratio, thereby**

**increasing the light attenuation and causing a larger difference in light attenuation between the sandy, vegetated sites and the siltier, unvegetated sites. Modifying the peak value as described above from 0.017 to 0.025 (47% increase; 0.025 value was used for a different mud-dominated estuary by Ganju et al., 2014) increased the median light attenuation at CB06 by 17% (from 1.35 to 1.58 m-1) and at CB11 by 11% (from 1.67 to 1.86). Therefore the spatial variability in light attenuation is likely robust and not confounded by variability in particle size."**

Determining the optimal sampling frequency is a science in itself and the results regarding the influence of sampling frequency (table 3) is interesting, but as presented it seems as an unfinished story that is not properly treated in the result and discussion section. This part should be either up- or down scaled/skipped.

**We believe that the concept that coarse temporal sampling masks spatial variability is an important one given the common daily sampling approach for dissolved oxygen in many impaired estuaries. We have expanded on this concept with examples from the literature in our revision, on P16, 19-23:**

**"Many estuary sampling programs, including "citizen science" efforts, often collect one daily sample of water quality parameters, including dissolved oxygen (Rheuban et al., 2016). Summers et al. (1997) demonstrated that month-long records of dissolved oxygen in a number of estuarine systems, resampled to replicate various non-continuous sampling programs, were not able to identify oxygen minima. This study expands on that finding by mimicking sampling programs within one large estuary, over an entire year."**

The present study do not examine gradients, so this term should be avoided (e.g. P1, L 13). Use "patchiness" or "spatial variability" instead.

**Agreed, we have revised throughout to "variability".**

Technical correction: Table 3: State which time period (yearly, season, other?), min, max and mean is covering. Include SD for the mean values. Figure 6: add x-axis titles

**We have revised to add time period to the caption, as well as standard deviation for the original data. We have added the x-axis (1/frequency) to Figure 6. Thank you for catching it.**

Mario Hoppema (Referee) mario.hoppema@awi.de

The manuscript presents an impressive data set for four locations in an estuary, including several useful variables/parameters. The data interpretation reveals some interesting results and conclusions, which are definitely worth publishing in Ocean Science.

**We appreciate the support and the willingness of the Editor to provide a comprehensive review.**

However, in some cases there seems to be over-interpretation, for example the difference in metabolic rates between vegetated and unvegetated sites.

**In addition to our response to Reviewer #1 above, we would indicate that the daily rates are markedly higher between CB3 and CB10 compare with CB6 – which is the main distinction we discussed between vegetated and unvegetated. We indicated in the revision that for mean period values, only the gross production ($P_g$) values between site CB03 and CB06 are statistically different.**

Also the data processing should be explained in somewhat more detail.

**See response above, we have added detail on the data processing on P7, 9-20.**

For the net ecosystem metabolism, sometimes NEM is used, for example in Figure 11, and sometimes
Pn, for example in Table 1. Please use only one single symbol or abbreviation throughout the
manuscript.

**We have corrected to "NEM" throughout.**

P6, L3 please define RBR D|Wave

**Revised to "RBR Virtuoso D|Wave pressure recorder".**

P6, L7 please define ADCP

**Revised to "acoustic Doppler current profiler (ADCP)".**

P9, L2 "across sites across habitats" Modify piece of text?

**Revised to "across sites and across habitats".**

P10, L15 I would recommend not to use NPQ as abbreviation, as it is only used twice; the reader possibly
has to search for it here.

**Revised to "non-photochemical quenching".**

P10, L19-20 "with the highest values consistently observed at CB11" I think one cannot state that,
because the record at CB11 is far from complete; certainly the word "consistently" is misleading here.

**Revised to "highest peak values over the overlapping period of record".**

P10, L20 "with lowest fDOM in the winter, possibly due to reduced biological activity" This does not
sound like a good explanation. fDOM is not high during maxima of chlorophyll, which seems to indicate
that it is not produced by biological activity. Moreover, the highest values seem to occur at site CB11,
which receives most riverine freshwater.

**This was meant to suggest that reduced activity in the surrounding watershed and water bodies
where DOM is produced, and then exported to the estuary with freshwater. We have clarified on P11,
11-13:**

> **"Concentrations were similar between the other sites, with lowest fDOM in the winter,
> possibly due to reduced dissolved organic material production in the surrounding watershed,
> and ultimately in freshwater runoff to the estuary."**

P11,L14-16"Light attenuation at site CB11,with its proximity to freshwater and nutrient sources, was
highest overall and more highly influenced by chlorophyll-a and fDOM than at other sites." This is hard
to believe when inspecting Figures 2-5. Moreover, there is much less data available for this site than for
the others.

**This is a fair comment, but the mean KdPAR is influenced strongly by the more frequent background conditions than the episodic values, which is why the mean ends up being larger at CB11, mostly caused by a higher background fDOM (Fig. 5). The partitioning of light attenuation between constituents is subject to error of the light model, but given that fDOM and chlorophyll-a are highest at that site (in terms of mean and peak values), it is expected that light attenuation would be highest there (despite lower turbidity than CB06). We have revised to indicate that this is only true during periods of overlap.**

P12. L13-14 "between May and October" My view of the figures says that this should be "between May and September".

**Revised.**

P12, L14 "during November to April" I was say "during November to March"

**Revised.**

P12, L1920 "but instances of net autotrophy (Pg > Rt) occurred nearly 70% of the time at the vegetated sites" When I view Figure 11, neither autotrophy nor heterotrophy are statistically significant. This should be mentioned here.

**It is a fair comment to suggest that we did not include statistical analyses to "prove" the existence of net autotrophy, but rather expressed the frequency of daily net autotrophic conditions. We have mentioned this on P13, L21-22:**

> **"Gross primary production and respiration were largely balanced across all sites, but the frequency of daily net autotrophy (Pg > Rt) was higher at the vegetated sites."**

P13, L10 at 1-7 day

**Revised.**

P13, L23-24 "The peak in spectral density was 30% higher at CB03 than CB10" This is really hard to see, if at all, in Fig. 6. Possibly the authors could add a comment to Figure 6 (Diss Oxygen) that the curve of CB3 lies under the one of CB10.

**We have added a comment in the caption.**

P17, L26 where the canonical C:N (I suggest to add canonical, since this is the well known Redfield ratio based on data from many locations)

**Revised.**

P19, L1-2 "In fact, modest net autotrophy prevailed during the summer season at vegetated sites but not at un-vegetated sites." This does not follow from the data in Figure 11, where NEM is around zero all through the year. See also comment above. Actually I am surprised by the uncertainty interval of NEM, which should have been formed frrom subtracting two larger terms. Because of this, I would expect it to be clearly larger and not smaller than the uncertainty of both of the terms.

**If you compare NEM to $P_g$ and $R_t$ on an absolute basis, we agree that NEM is near zero. But our statement simply refers to the mean monthly value of NEM, which although much smaller than $P_g$ and $R_t$, is still greater than zero at the vegetated sites during some seasons, which we define as**

**autotrophic. The error bars around NEM represent the standard deviation of the monthly mean estimates of NEM, which the reviewer is correct in that they are derived as the difference between $P_g$ and $R_t$. But these difference calculations are performed on a daily basis, so the error bars represent the total variation of all ~30 daily Pg, Rt, and NEM rates and thus do not represent uncertainty carried over from each individual NEM calculation.**

P26, L18 The usual abbreviation is: Limnol. Oceanogr.

P26, L22 Use full journal name: Biogeosciences

P27, L2 The usual abbreviation is: Estuar. Coast. Shelf Sci.,

P27, L12 Please give more info on this publication; report, website?

P27, L14 dito

P27, L16 The usual abbreviation is: Limnol. Oceanogr.

P27, L18 Ocean Mod.

P27, L20 Please add page numbers

**We have corrected all reference errors.**

Table 1 Please spell out standard deviation

**Revised.**

Figure 1 Please also define ADCP and CBWS .

**Revised.**

Figure 6 I think CB10 in red under the figure panels must be CB11 (unvegetated) Please indicate panels (a) – (d), and then also in the caption. Please explain the x-axis; in the main text the authors talk about "peak at 12.4 h", but this is not reflected in the figure.

**Revised, the 12.4 peak, at ~ 0.5 days, is now noted in the caption.**

Figure 7 It is not clear to me how the oxygen saturation could be 130% in the middle of winter (e.g. at CB11).

**The periods where oxygen saturation exceed 130% do not truly occur in winter, which we would define as mid-December to mid- March. The exception to this is during a brief period in February and early March at CB 11, where chlorophyll-a exceeded 25 ug/L indicating that phytoplankton biomass was high and presumably, primary production rates.**

Figure 9 Please indicate panels (a) – (d), and then also in the caption. Please explain what the bottom structures in the panels mean, and why the data are shown despite these structures.

**Revised; the shaded areas are zones of the parameter space that are not statistically robust. We have clarified this in the caption.**

[revised manuscript text omitted]